# Impact of Exogenous *Lactiplantibacillus plantarum* on the Gut Microbiome of Hematopoietic Stem Cell Transplantation Patients Colonized by Multidrug-Resistant Bacteria: An Observational Study

**DOI:** 10.3390/antibiotics13111010

**Published:** 2024-10-28

**Authors:** Bruna D. G. C. Moraes, Roberta C. R. Martins, Joyce Vanessa da Silva Fonseca, Lucas A. M. Franco, Gaspar C. O. Pereira, Thais F. Bartelli, Marina F. Cortes, Nazareno Scaccia, Carolina F. Santos, Priscila T. Musqueira, Leonardo J. Otuyama, Victor S. Pylro, Livia Mariano, Vanderson Rocha, Steven S. Witkin, Ester Sabino, Thais Guimaraes, Silvia Figueiredo Costa

**Affiliations:** 1LIM-46 and LIM49, Department of Infectious Diseases, Faculdade de Medicina, Universidade de Sao Paulo, Av. Dr. Enéas Carvalho de Aguiar, 470 Jardim America, São Paulo 05403-000, SP, Brazil; brudelguerra@yahoo.com.br (B.D.G.C.M.); betacristina@gmail.com (R.C.R.M.); joycevanesf@gmail.com (J.V.d.S.F.); lucasamfranco@hotmail.com (L.A.M.F.); gasparcamilo227@gmail.com (G.C.O.P.); marinafarrel@yahoo.com.br (M.F.C.); nazareno.scaccia@gmail.com (N.S.); sabinoec@gmail.com (E.S.); 2Department of Hematology, Hemotherapy and Cell Therapy Service, Faculdade de Medicina, Universidade de Sao Paulo, Sao Paulo 01246-000, SP, Brazil; carolina.ferreira@hc.fm.usp.br (C.F.S.); priscila.musqueira@hc.fm.usp.br (P.T.M.); leonardo.otuyama@hc.fm.usp.br (L.J.O.); livia.caroline@hc.fm.usp.br (L.M.); vanderson.rocha@hc.fm.usp.br (V.R.); 3Centro Internacional de Pesquisa, CIPE, A.C.Camargo Cancer Center, Rua Taguá 440, São Paulo 01508-010, SP, Brazil; thais.bartelli@accamargo.org.br; 4Department of Biology, Federal University of Lavras, Lavras 37200-000, MG, Brazil; victor.pylro@ufla.br; 5Microbial Ecology and Bioinformatics, Biology Department, Federal University of Lavras—UFLA, Lavras 37200-900, MG, Brazil; 6Division of Immunology and Infectious Diseases, Weill Cornell Medicine of Cornell University, New York, NY 10065, USA; switkin@med.cornell.edu; 7Hospital das Clinicas da Faculdade de Medicina da Universidade de São Paulo, São Paulo 01246-000, SP, Brazil; tguimaraes@terra.com.br

**Keywords:** *Lactiplantibacillus plantarum*, gut microbiome, gut colonization by multidrug-resistant organisms, hematopoietic stem cell transplantation

## Abstract

**Background:** *Lactiplantibacillus plantarum* can inhibit the growth of multidrug-resistant organisms (MDROs) and modulate the gut microbiome. However, data on hematopoietic stem cell transplantation (HSCT) are scarce. **Aim:** In an observational study, we assessed the impact of *L. plantarum* on the modulation of the gut microbiome in HSCT patients colonized by MDROs. **Methods:** Participants were allocated to an intervention group (IG = 22) who received capsules of *L. plantarum* (5 × 10^9^ CFU) twice per day until the onset of neutropenia or a control group (CG = 20). The V4 region of the 16S bacterial rRNA gene was sequenced in 87 stool samples from a subset of 33 patients (IG = 20 and CG = 13). The Phylogenetic Investigation of Communities by Reconstruction of Unobserved States (PICRUSt2) program was used to predict metagenome functions. **Results:** *L. plantarum* demonstrated an average 86% (±11%) drug-target engagement at 43 (±29) days of consumption and was deemed safe, well-tolerated, and associated with an increase in the abundance of the Lactobacillales (*p* < 0.05). A significant increase in *Lactococcus* and a reduction in *Turicibacter* (*p* < 0.05) were identified on the second week of *L. plantarum* use. Although *Enterococcus* abundance had a greater rise in the CG (*p* = 0.07), there were no significant differences concerning the Gram-negative MDROs. No serious adverse effects were reported in the IG. We observed a greater, non-significant pyruvate fermentation to propanoate I (*p* = 0.193) relative abundance in the IG compared with the CG. *L. plantarum* use was safe and tolerable by HSCT patients. **Conclusions:** While *L. plantarum* is safe and may impact *Enterococcus* and *Turicibacter* abundance, it showed no impact on Gram-negative MDRO abundance in HSCT patients.

## 1. Introduction

Gut colonization by multidrug-resistant organisms (MDROs) has been reported as an independent risk factor for bloodstream infection (BSI) with high mortality among allogeneic hematopoietic stem cell transplant (HSCT) patients [1,2]. Strategies for reducing MDRO colonization and infection are challenging, with most studies focusing on the use of antimicrobial agents [3].

Gut microbiome diversity has been associated with diet, antibiotic use, and chemotherapy in HSCT patients. Higher diversity of gut microbiome at the time of neutrophil engraftment was associated with lower mortality and graft-versus-host diseases in allogeneic HSCT [4]. The use of anti-anaerobic antibiotics decreased gut microbiome diversity in allogeneic HSCT patients [5]. Moreover, the impact of using antibiotics with different anaerobic spectra of activity and fluoroquinolone prophylaxis on gut microbiome is controversial. A recent study showed no significant impact of ending ciprofloxacin prophylaxis on microbiome diversity [6].

Previous gut colonization has been associated with barrier BSI in neutropenic HSCT. At our center, Ferreira et al. evaluated 232 HSCT patients and demonstrated an MDRO colonization rate of 30.5%. Bloodstream infection (BSI) occurred in 20% of the MDRO-positive group. The authors observed, by multivariate analysis, that prior gut colonization by Gram-negative MDROs was a risk factor for this outcome [1]. Previous studies also observed that the increase in *Enterococcus* in the microbiome during the engraftment period is associated with BSI during the HSCT [7]. Few studies, however, have attempted to modulate the presence of MDROs in the intestinal microbiome by focusing on the individual’s diet or the use of prebiotics, probiotics, and symbiotics [8,9,10,11]. Moreover, this has rarely been reported in patients undergoing HSCT. Probiotics are live microorganisms that can provide benefit to the host if present in adequate doses [12]. Usually, probiotics are not offered to immunocompromised patients, like those undergoing HSCT, due to a risk of translocation and infection. To date, only one study demonstrated the safety and feasibility of using a specific probiotic, *Lactiplantibacillus plantarum*, in pediatric neutropenic patients undergoing HSCT [13].

The presence of *Lactiplantibacillus* spp. in the fecal microbiota of hospitalized subjects has been associated with a lack of MDRO acquisition, indicating a potential protective role for these bacteria in HSCT [14]. Studies in vitro have shown that *L. plantarum* can inhibit Gram-positive and Gram-negative bacteria due to bacteriocins and that acetic acid produced by *L. plantarum* can inhibit the growth of carbapenem-resistant *K. pneumonia* [15,16,17]. *Lactiplantibacillus plantarum* CIRM653 can impair *K. pneumoniae* preformed biofilm and *L. plantarum*-derived extracellular vesicles, modulating host responses to vancomycin-resistant *Enterococcus faecium* (VRE) [17,18].

*Lactiplantibacillus plantarum* can have modulatory effects on the intestine in healthy individuals by reducing proteins such as matrix metallopeptidases and zonulin [19]. However, a recent study using whole-genome sequencing described a cluster of BSI due to *Lactiplantibacillus* in pediatric patients undergoing HSCT who received probiotics [20]. Thus, more studies on the safe use of probiotics in HSCT are needed.

In the present study, we assessed the use of *L. plantarum* (intervention) as a probiotic in patients undergoing HSCT who were colonized by MDROs. We hypothesized that this intervention would reduce MDRO colonization and infection compared with an untreated control group. Additionally, we assessed the impact of the intervention on the gut microbiome and the safety and adverse effects of *L. plantarum* usage in HSCT patients.

## 2. Results

*Patient characteristics*. A total of 55 candidates for HSCT were selected for the study, with nine excluded for not having collected the baseline stool sample (see Figure 1). The final cohort included 42 participants: 20 in the CG and 22 in the IG. However, stool microbiome was not conducted for all participants. In the IG, two participants did not undergo HSCT due to disease refractoriness, and one participant was yet to undergo HSCT by the end of this study. Therefore, the analysis of MDROs and the modulation of the fecal microbiome following *L. plantarum* ingestion included 22 participants in the IG and 13 in the CG. Thirty-two patients enrolled in the study were autologous HSCTs, most of the patients received myeloablative treatment, and lymphoma and gammopathy were the most frequent underlying diseases. The mean duration of neutropenia was 13 days (10–20 days) in the IG and 13 days (10–18 days) in CG.

The patient and HSCT characteristics, along with gastrointestinal toxicities observed during HSCT, are presented in Table 1.

### 2.1. Lactiplantibacillus plantarum

The analysis of *L. plantarum* capsules with next-generation sequencing (NGS) showed an abundance of *Lactobacillaceae*. Comparing the most abundant amplicon sequence variants (ASVs) in the samples with GenBank (NCBI) revealed a 99% identity with *L. plantarum*. Additionally, Maldi-TOF analysis of the *L. plantarum* capsules showed high correspondence with *L. plantarum*. However, bacterial counts in the culture medium demonstrated a deficit in the expected quantity of *L. plantarum* colony-forming units (CFUs) in the capsules, measuring 6 × 10^8^ ± 3.45 × 10^8^ CFU instead of the requested 5 × 10^9^ CFU. Therefore, we adjusted the number of prescribed probiotic capsules for the IG from one capsule to two twice per day.

The *L. plantarum* capsules were consumed at an average rate of 86% (±11%) by day 43 (±29). The mean duration of *L. plantarum* use was 48 days (range 13 to 108 days) in the IG; among individuals that developed BSI, it was 19 days (range 13–47 days). Three subjects reported mild (grade 1 in CTCAE) flatulence, while three others reported moderate (grade 2) flatulence. Of the 16 patients who reported Bristol Scale Stool type 3 before *L. plantarum* ingestion, only one remained at type 3 after its use, while the other 15 subsequently reported type 4 stools.

### 2.2. MDRO Colonization

All patients included in the study were colonized by MDROs. In the evaluation before the onset of HSCT (median 6 days, IQR 7), 3/20 (15.0%) individuals in the CG continued to be colonized by vancomycin-resistant Enterococcus (VRE). In the IG, 6/22 patients (27.0%) continued to be colonized, three by VRE, and the other three by Carbapenem-resistant *Enterobacteriaceae*, with a median consumption of *L. plantarum* of 17 days (IQR 7) and 47 days (IQR 36), respectively. The median number of days between the beginning of *Lactiplantibacillus* and the decolonization evaluation was 19 days (IQR 47). There was no new colonization by other MDROs during the follow-up period for both groups.

### 2.3. Infection

Febrile neutropenia occurred in 16/20 (80.0%) individuals in the CG and 10/20 (52.6%) in the IG (*p* = 0.14). In the IG, seven patients developed MBI-LCBI, and five patients had *Klebsiella pneumonia* infection. Among the five IG patients who developed MBI-LCBI due to *Klebsiella pneumonia*, three had been colonized by Carbapenem-resistant *K. pneumonia* (Table 2).

### 2.4. Stool Microbiome Analysis

Stool samples were collected at various time points from the CG (baseline n = 13 and on week grafting n = 10) and IG (baseline n = 23, second week of *L. plantarum* use n = 22, at the time of neutropenia n = 11, one week before infection n = 4, and grafting n = 10). The presence of *Lactobacillus* was assessed at different time periods for the IG. Microbiome sequencing data are available on Bioproject (NCBI—National Center for Biotechnology Information, U.S. National Library of Medicine) under accession number PRJNA724885.

Overall, there was a reduction in alpha diversity (Observed ASVs, Shannon, and Simpson indices) for the IG and CG patients across different periods, irrespective of *L. plantarum* use, with significant values, especially between baseline and the second week versus neutropenia and grafting (Wilcoxon *p* < 0.05 and *p* < 0.01). Additionally, intense intraindividual variability in alpha diversity between these points was found in both IG and CG. Alpha diversity by Faith’s index decreased between baseline and grafting in both groups (IG *p* = 0.0026; CG *p* = 0.018) (Figure 2A(IG and CG),B(IG)).

When analyzing the composition of both IG and CG, we observed a significant change in their bacteriome, both qualitatively and quantitatively, between baseline and grafting (ANOSIM—Bray–Curtis, weighted and unweighted UniFrac *p* < 0.05) (see Figure 3).

The bacterial composition of both groups was compared on these two points, (IG vs. CG at baseline—ANOSIM UniFrac *p* = 0.002, R = 0.298; Bray–Curtis *p* = 0.007, R = 0.251; wunifrac *p* = 0.004, R = 0.21); (CG vs. IG at grafting—ANOSIM UniFrac *p* = 0.001, R = 0.435; Bray–Curtis *p* = 0.004, R = 0.308; wunifrac *p* = 0.008, R = 0.279).

At baseline, at least 50.9% of bacteria were predominantly from the phyla Bacteroidetes, followed by Firmicutes (38.3%) and Proteobacteria (4.5%). These phyla were still predominant at grafting for both IG and CG. Proteobacteria increased in both groups between baseline and grafting (*p* = 0.009) without significant differences between groups (*p* = 0.41).

The detection of Firmicutes and Bacteroidetes phyla was comparable in the IG and CG at baseline (*p* = 0.18 and *p* = 0.72, respectively) and following grafting (*p* = 0.50 and *p* = 0.14, respectively).

Stool samples from four IG patients who developed BSI were sequenced 1 week before the infection event. One patient with *K. pneumoniae* had a gut microbiome dominated by *Enterobacteriaceae*. Two other patients showed an increase in the *Bacteroides* genus, one with BSI by *E. coli* ESBL-positive and the other by carbapenem-resistant *K. pneumonia*. 

The consumption of *L. plantarum* was associated with an increased presence of *Lactobacillales* (*p* < 0.05) in the second-week period of its use. Differences between baseline and the second week measured by LefSe with alpha values for the factorial Kruskal–Wallis test *p* < 0.02 and LDA > 1 showed an increase in *Lactococcus*, accompanied by a decrease in *Turicibacter* in the IG (n = 22) (Figure 4).

The Lactobacillaceae family was identified as differentially abundant in the second week of *L. plantarum* use in the IG using Kruskal–Wallis factorial test; alpha value of *p* < 0.01 and an LDA score > 3.

Although no differentially abundant bacteria were identified between the IG and CG, both groups presented a decrease in the presence of *Roseburia* and *Coprococcus* when baseline and grafting were compared (Figure 5).

We observed an increase in the abundance of *Enterococcus* in the CG, with a trend toward significance compared with the IG (*p* = 0.07) in paired sample analysis, considering patients with stool collected in both periods (baseline and grafting) (Figure 6).

### 2.5. Metagenome Functions

The PICRUSt2 program was used to predict the metagenome functions of the gut microbiome of HSCT patients. We observed a greater, though statistically non-significant, relative abundance of pyruvate fermentation to propanoate I (*p* = 0.193) in the IG (yes-Lac) compared to the CG (no-lac) (Figure 7).

## 3. Discussion

Of note, *L. plantarum* demonstrated safety and tolerability before the onset of neutropenia in all individuals undergoing HSCT, and it may potentially reduce fecal levels of the *Enterococcus* genus in HSCT patients. There was a decrease in the presence of *Roseburia* and *Coprococcus* in the gut microbiome during the grafting period in both IG and CG. The consumption of *L. plantarum* was associated with a significant increase in *Lactobacillales* and a decrease in *Turicibacter* in the gut microbiome of HSCT patients during the second week of usage. None of the patients in the intervention group experienced BSI caused by *L. plantarum* or severe adverse events.

While a significant reduction in MDRO colonization or infection following *L. plantarum* ingestion compared to the control group was not observed, all infections in the intervention group were MBI-LCBI, which occurred after the cessation of *L. plantarum* use. Despite individuals in the IG being more frequently colonized by three MDROs (VRE, carbapenem-resistant *K. pneumoniae*, and carbapenem-resistant *A. baumannii*), polymicrobial MDRO-BSI and carbapenem-resistant *A. baumannii*-BSI were not identified in the intervention group, suggesting the potential benefits of *L. plantarum* use.

Data on the impact of *Lactiplantibacillus* on the prevention of MDRO colonization are controversial and dependent on many variables, such as species, antibiotic use, and patient population [8]. A recent randomized controlled study involving 120 patients treated with amoxicillin-clavulanate for 10 days compared the effects with a 30-day treatment with placebo *Saccharomyces boulardii* and a probiotic mixture containing *Saccharomyces boulardii, Lactiplantibacillus acidophilus* CFM, *Lactiplantibacillus paracasei*, *Bifidobacterium lactis* 4, and *Bifidobacterium lactis*. It was noted that treatment with the probiotic mixture led to a significant decline in colonization by *Pseudomonas* (*p* = 0.041) and AmpC-producing enterobacteria [11]. On the other hand, another prospective, double-blinded, randomized controlled trial found that *Lactiplantibacillus rhamnosus* did not prevent MDRO colonization among patients receiving broad-spectrum antibiotics [21].

The *Enterobacteriaceae* family, mainly *K. pneumoniae*, was responsible for the highest number of infections in patients in the IG. Intestinal domination by *Enterobacterales* was observed in only one patient who developed BSI. Interestingly, the IG developed MBI-LCBI during the neutropenia period after *L. plantarum* was discontinued. In samples from these individuals, members of the *Lactobacillales* order were not found.

Most patients enrolled in our study underwent autologous HSCT. Few studies have addressed the gut microbiome in autologous HSCT. A recent study observed loss of alpha diversity at the engraftment timepoint driven by decreases in *Blautia*, *Ruminococcus*, and *Faecalibacterium* genera in 33 patients with multiple myeloma undergoing autologous HSCT [22]. We observed a significant reduction in alpha diversity in the gut microbiome, especially between baseline and the second week versus neutropenia and grafting. Additionally, intense intra-individual variability in alpha diversity between the time points was found, as previously shown by Taur et al. [7]. Interestingly, there was a reduction in the uniformity of the species observed between baseline and grafting in both groups, which was slightly smaller in CG than in IG. At baseline, there was a predominance of the phyla Bacteroidetes, followed by Firmicutes and Proteobacteria, which are considered a healthier gut microbiome. On the other hand, there was a decrease in the presence of *Roseburia* and *Coprococcus*, short-chain fatty acid producers in the gut microbiome, during the grafting period in both groups, IG and CG, similar to what was described in allogeneic HSCT patients [23].

Although Enterococcus abundance had a greater increase in the CG, there were no significant differences between groups, possibly due to the cessation of *L. plantarum* during neutropenia and the small sample size. Previous studies have observed that the increase in Enterococcus in the microbiome during the engraftment period is associated with BSI during HSCT [24,25,26,27]. Thus, the lower Enterococcus abundance found in the IG might be advantageous. In addition, an increase in *Lactococcus* and a decrease in *Turicibacter* abundance was noted on the second week of *L. plantarum* use in the IG. The increase in *Lactococcus* may be worthy of note because this genus expresses an antimicrobial peptide involved in human gut homeostasis and may reduce VRE [24,25,26]. *Turicibacter* is believed to contribute to host metabolic and homeostatic mechanisms [27]. Thus, it seems that *L. plantarum* could be useful in decreasing *Enterococcus* abundance, potentially reducing VRE, and aiding in the management of HSCT patients in settings with a high prevalence of VRE.

Various methods have been developed to predict functions from 16S rRNA sequence data, including PICRUSt2, which was applied in the present study [28]. We observed a non-significant increase in metabolites, such as pyruvate fermentation to propanoate I (*p* = 0.193) relative abundance in the IG. Bacteria from the *Firmicutis* phylum, such as f_*Lactobacillaceae* and *g_Lactobacillus*, have been associated with pyruvate fermentation to propanoate I in Parkison’s disease [29]. However, the role of *L. plantarum* in metagenomic functions in the gut microbiome of HSCT patients needs further clarification.

Of note, *L. plantarum* had an average 86% (±11%) drug-target engagement after 43 (±29) days of consumption. The intervention proved to be safe, tolerable, and significantly associated with an increase in the abundance of Lactobacillales in the present study. None of the patients in the IG experienced BSI caused by *L. plantarum* or serious adverse events. This aligns with findings from two recent studies involving HSCT patients, supporting the safety of probiotics in this population [13,20]. These studies highlight that the organisms found in over-the-counter probiotics are rarely implicated in BSI during the pre-engraftment period, indicating the feasibility of administering probiotic *L. plantarum* to children undergoing HSCT without associated BSI or serious adverse events [13].

This study has several limitations: it is an observational study without a control group that received a placebo, and the duration of *L. plantarum* consumption varied among patients based on their availability and clinical condition for admission to HSCT. Additionally, the sample size was small and not controlled for clinical features that could potentially influence the results.

## 4. Materials and Methods

### 4.1. Study Setting

This study was conducted at the Cell Therapy Clinical Unit (CTCU) at the Hospital das Clinicas, Faculdade de Medicina, Universidade de São Paulo, Brazil. The CTCU is a 10-room ward for adult patients, with double bedrooms for autologous transplants and single bedrooms for allogeneic transplants. Piperacillin-tazobactam was the empirical antibiotic used during febrile neutropenia.

### 4.2. Study Design

This exploratory clinical investigation included patients seen from November 2017 to May 2020 who were colonized by MDROs prior to their HSCT. We utilized a convenience sample of patients treated with *L. plantarum* (IG) and an untreated control group (CG). The inclusion criteria were: oncohematological disease and indication for HSCT, being over 18 years old with no history of previous gastrointestinal tract surgery, and colonization by MDROs identified by a positive surveillance culture (SC) prior to HSCT. The study protocol followed Helsinki’s declaration and was approved by the Institute’s ethical committee (CAPPesq approval 2.126.478), registered on the Registro Brasileiro de Ensaios Clínicos (ReBEC, Brazilian Registry of Clinical Trials, number RBR-2ztbwgr), and all participants voluntarily signed the written informed consent form.

Demographic and clinical variables were assessed, such as age, gender, underlying diseases, HSCT, mucositis, weight, height, body mass index (BMI), and classification of nutritional status by WHO or PAHO criteria at the time of admission to the study. Feces consistency was determined using the Bristol fecal scale prior to and following the consumption of *L. plantarum*. Gastrointestinal toxicities (mucositis and diarrhea) were measured by Common Terminology Criteria for Adverse Events (CTCAE—Version 5.0) [30]. Neutrophil recovery was defined as the third day of an absolute neutrophil count > 500 neutrophils mm^−3^.

### 4.3. Lactiplantibacillus plantarum Administration

*Lactiplantibacillus plantarum, G18* lineage, was administrated as a probiotic to the treatment group at a dose of 5 × 10^9^ colony forming units (CFU) twice daily in the form of gastric release capsules produced in a compounding pharmacy. The HSCT pharmacy supervised capsule dispensing and assessed drug-target engagement quantification. Consumption was terminated in cases of HSCT neutropenia. All symptoms in those taking *L. plantarum* were assessed according to CTCAE protocol [30]. The safety of the use of *L. plantarum* was also evaluated by observing the development of BSI by *L. plantarum* up to day +60 after its use. The content of *L. plantarum* capsules was analyzed by 16S rRNA next-generation sequencing (NGS), growth in *Lactiplantibacillus* culture medium (MRS Agar), and mass spectrometry (Maldi-TOF-Buker-Bremen, Germany).

### 4.4. Definitions

MDROs: Vancomycin-resistant *Enterococcus*; *Enterobacteria* resistant to carbapenems and/or colistin, producing ESBL (enzymes produced by certain bacteria that are able to hydrolyze extended-spectrum cephalosporin); Gram-negative non-fermenting bacteria (*Pseudomonas aeruginosa* and *Acinetobacter baumannii*) resistant to carbapenems and/or colistin [31].

Colonization: Colonization was assessed by SC using rectal swab samples and qPCR analysis for the main resistance genes: *bla*_VIM_—metallo-beta-lactamase VIM, *bla*_SPM_—metallo-beta-lactamase SPM, *bla*_KPC_—carbapenem-hydrolyzing beta-lactamase KPC, *mcr-1*—Phosphoethanolamine-lipid A transferase MCR-1, *bla*_OXA-23_—carbapenem-hydrolyzing beta-lactamase OXA-23, *bla*_OXA-48_—carbapenem-hydrolyzing beta-lactamase OXA-48, *bla*_OXA-143_—carbapenem-hydrolyzing beta-lactamase OXA-143, *van*A—vancomycin-resistance protein VanA, *van*B—vancomycin-resistance protein VanB. The DNA sequences of oligonucleotides were used in qPCR as previously described [31,32,33,34].

Infections were defined using the criteria of the Centers for Disease Control and Prevention (CDC) [35]: central line-associated bloodstream infections (CLABSIs) were characterized as laboratory-confirmed bloodstream infections where an eligible BSI organism is identified on the day of the event or the preceding day. Mucosal barrier injury–laboratory-confirmed bloodstream infection (MBI-LCBI) characterizes LCBI in patients immunosuppressed by microbiological translocation of the gastrointestinal tract due to persistent neutropenia (neutrophils < 500 cells/mm^3^), diarrheal episodes (1 L or more of diarrhea in 24 h), or graft-versus-host disease (GVHD) in allogeneic HSCT patients, within seven days of positive blood culture. Decolonization was defined as the SC changing from positive to negative after the intervention period.

### 4.5. Sample Collection

Stool samples were collected in an outpatient setting according to hospital protocol. Surveillance cultures were performed weekly in inpatient settings for HSCT patients. Bacterial cultures on selective media were initiated, followed by qPCR targeting key resistance genes. Specifically, a multiplex PCR for detecting genes related to *Metallo-β-lactamase* resistance in *P. aeruginosa* (*IMP*, *VIM*, and *SPM*), oxacillinase resistance in *A. baumanii* (*OXA-23* and *OXA-143*), genes related to *Van A* and *Van B* resistance, and for detecting genes related to *bla*_KPC_ antimicrobial resistance [31,32,33,34]. Stool samples were obtained prospectively in sterile bottles containing guanidine and stored at −20 °C until DNA extraction, as previously described [36]. The first collection, termed ‘baseline’, occurred before probiotic consumption. Patients were instructed to date and store samples in a freezer until delivery to the researcher, or they were collected on the day of delivery. Collections were performed weekly after initiating the consumption of *L. plantarum* and continued until the grafting procedure.

### 4.6. Microbiome Processing

The composition of the prokaryotic communities was determined based on partial 16S rRNA (V4 region) sequences directly amplified using a bacterial/archaeal primer set 515F/806R [33] from each DNA sample. PCR amplification, library preparation, and sequencing followed the procedures described by Ribeiro et al. [36]. Briefly, approximately 0.25 g of feces was used for DNA isolation using the DNeasy PowerSoil Kit (Qiagen, Germantown, MD, USA), following the manufacturer’s instructions. The template preparation was performed by the Ion Chef System (Thermo Fisher Scientific, Waltham, MA, USA), using the Ion PGM Hi-Q View Chef Kit. Sequencing was performed in the Ion Personal Genome Machine (PGM), using the Ion PGM Hi-Q Sequencing Kit and the Ion 318 Chip v2, following the manufacturer’s instructions (Thermo Fisher Scientific, Waltham, MA, USA). Samples with less than 85.000 reads were re-sequenced.

### 4.7. Stool Microbiome Analysis

The 16S rRNA gene data pre-processing and diversity estimates were performed using Quantitative Insights Into Microbial Ecology (QIIME2) version 2019.10 [37]. Demultiplexed sequence data were denoised with DADA2 (via q2-dada2) with the default parameters: 260 bp in length and an average quality Phred score of ≥30 to generate the amplicon sequence variants (ASVs) [38,39]. A phylogenetic tree was built by inserting the sequences into the Greengenes 13_8 reference tree using the q2-fragment-insertion plugin [40], which uses the SATé-enabled phylogenetic placement insertion method [36]. Alpha-diversity metrics (number of observed ASVs, Pielou’s evenness, Shannon diversity, and Faith’s phylogenetic diversity) and beta-diversity metrics (Bray–Curtis and unweighted and weighted UniFrac) were estimated using q2-diversity after samples were rarefied to 32,000 sequences per sample [41,42]. The rarefaction curves for the samples reached the plateau, indicating that there was a good representation of the microbial community. The principal coordinates analysis (PCoA) plot for each of the beta-diversity metrics was generated using the phyloseq package as implemented in R [40]. The ASVs were taxonomically classified using the q2-feature-classifier [43] (naive Bayes classifier) against Greengenes 13_8 99% OTU reference sequences [44,45,46,47]. Metagenomic functions were analyzed using Package: ggpicrust2, Type: Package, Title: Make ‘PICRUSt2’ Output Analysis and Visualization Easier, Version: 1.7.2, R version 4.3.1 (16 June 2023) [48].

### 4.8. Statistical Analyses

A database containing all demographic and clinical variables was built using the Collaborative software Airtable-version 1 (https://www.airtable.com/) (San Francisco, CA, USA). Categorical variables were described as counts and proportions. Variables with a normal and asymmetric distribution were described as mean (range). Normality was evaluated with a visual inspection of histograms and the Shapiro–Wilk test. Student’s *t*-test was used for the comparison of normal variables, while the Wilcoxon test was used for non-normal variables. Categorical variables were compared using the chi-square and Fisher exact tests (OR and 95% CI were described for categorical variables). Microbiota community compositions were summarized proportionally at various taxonomy levels, including species, genus, family, order, class, and phylum ranks. For all analyses, significance was determined as *p* < 0.05. The Kruskal–Wallis test was performed to explore differences in alpha-diversity metrics. Differences in community composition (beta-diversity) were assessed using non-parametric analysis of similarities (ANOSIM) tests. To determine the features most likely to explain differences between periods and groups, we employed the algorithm linear discriminant analysis effect size—LEfSe—version 1.1.2 [45]. Wilcoxon rank-sum tests and Wilcoxon signed rank tests, adjusted by Bonferroni correction, were used to compare continuous microbiota features among groups and time periods. Fisher’s exact test was applied to compare categorical decolonization data. All these analyses were performed using R, version 3.6.

## 5. Conclusions

In conclusion, the ingestion of *L. plantarum* capsules was deemed safe and well-tolerated. While there may be an impact on Enterococcus abundance and a reduction in *Turicibacter*, no significant impact on Gram-negative MDRO abundance in HSCT patients was observed. Further studies with more individuals are necessary to confirm our findings.

## Figures and Tables

**Figure 1 antibiotics-13-01010-f001:**
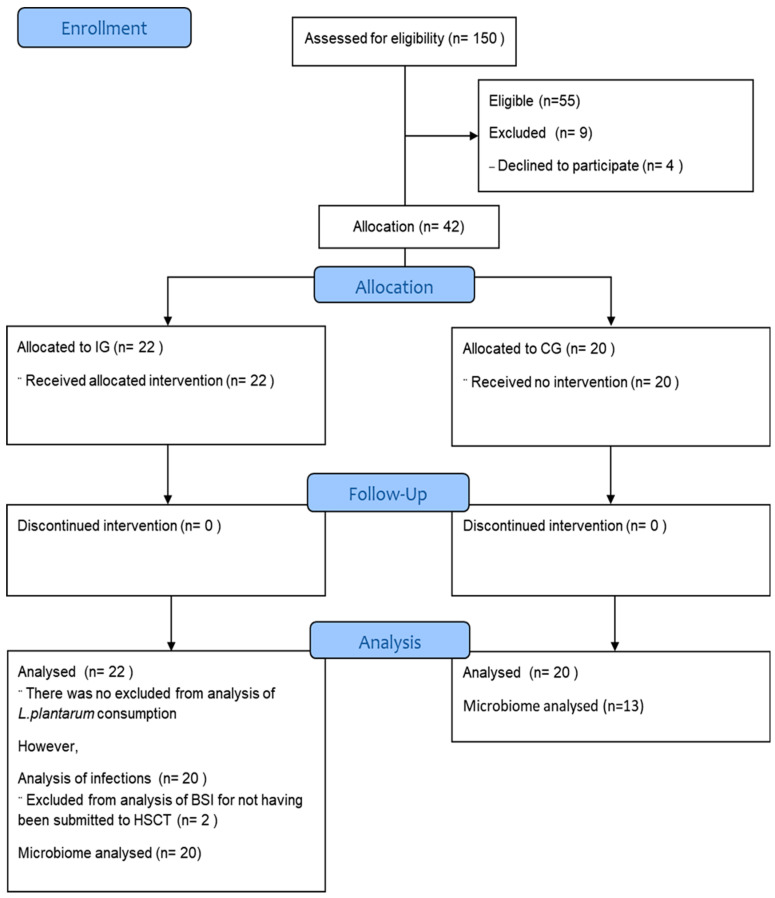
The CONSORT flow diagram illustrates the progress through the phases of a parallel trial, depicting the study groups and controls. Exclusion criteria: Patients who did not collect stool samples after initiation of HSCT conditioning.

**Figure 2 antibiotics-13-01010-f002:**
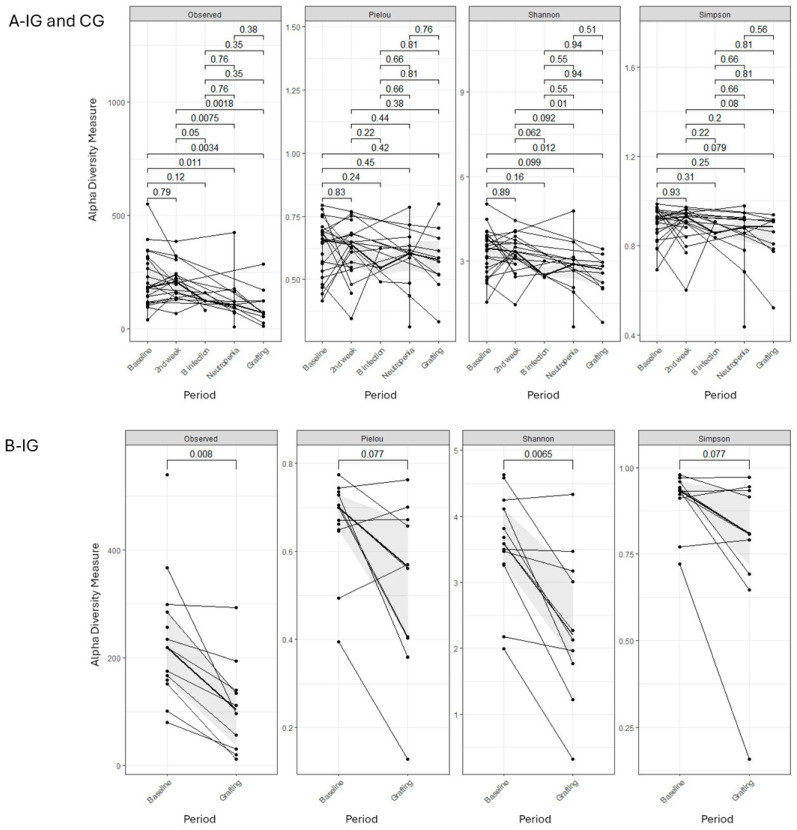
Alteration in Alpha diversity by Observed ASV, Shannon, Simpson, and Pielou’s evenness index among HSCT patients between periods for both IG and CG ((**A**)-IG and CG and (**B**)-IG). The alpha diversity indexes were tested for differences using Wilcoxon rank sum test. The continuous line represents the median and the shaded region the 25 and 75 percentiles.

**Figure 3 antibiotics-13-01010-f003:**
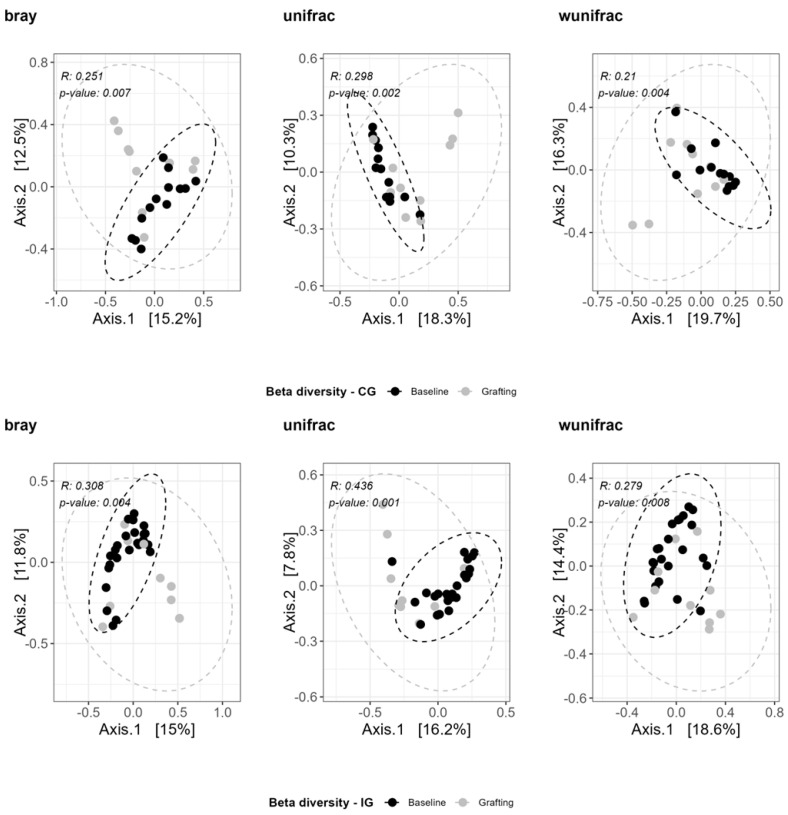
Bacterial beta diversity measures on stool samples of HSCT patients enrolled in the IG and CG (ANOSIM *p* < 0.05 significant) at baseline and grafting.

**Figure 4 antibiotics-13-01010-f004:**
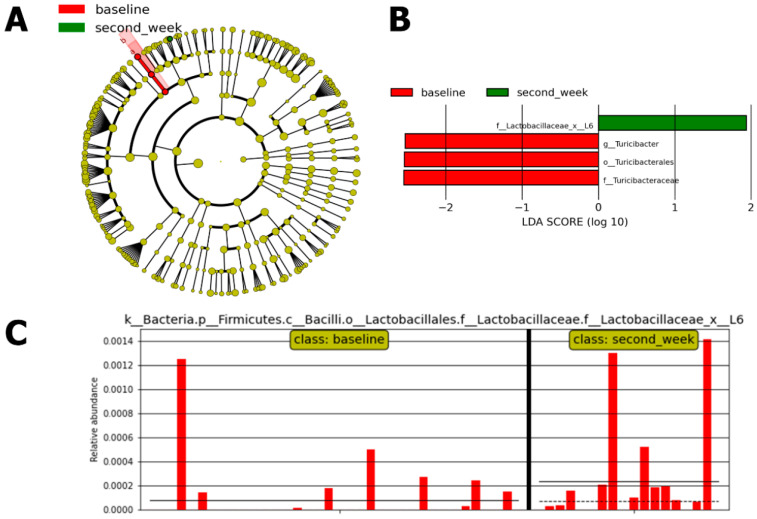
Effect size of linear discriminant analysis (LefSe). (**A**) Cladogram representing taxonomic differences between baseline and *L. plantarum* consumption in the second week in the Intervention Group (n = 22). (**B**) LefSe demonstrating enrichment of Lactobacillaceae family abundance after two weeks of *L. plantarum*. (**C**) Histogram of all samples displaying linear discriminant analysis (LDA) scores for differentially abundant features between baseline and the second week of *L. plantarum* use (*p* < 0.02 and LDA score > 1 were considered significant and are presented here with notation indicating the corresponding taxonomy [K = Kingdom; p = phylum; g = genus; f = family; o = order. Solid line: mean; dashed line: median]).

**Figure 5 antibiotics-13-01010-f005:**
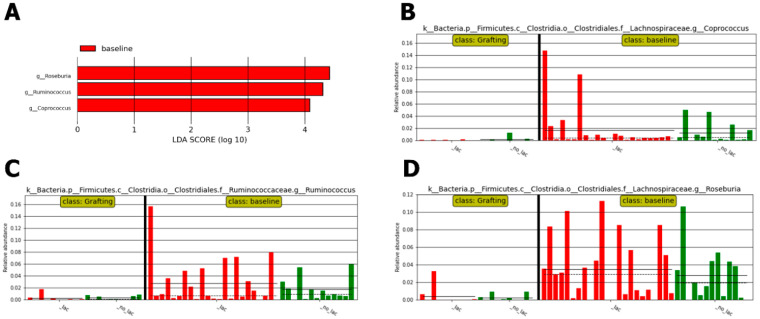
Potential biomarkers were defined by the effect size of linear discriminant analysis (LefSe) between baseline and engraftment periods. (**A**) Histogram of linear discriminant analysis (LDA) scores for differentially abundant features between baseline and engraftment (*p* < 0.01; LDA score > 3 was considered significant, solid line mean and dashed line median]). (**B**–**D**) LefSe demonstrates a decreased presence of *Coprococcus*, *Ruminococcus*, and *Roseburia* during the engraftment period compared to baseline.

**Figure 6 antibiotics-13-01010-f006:**
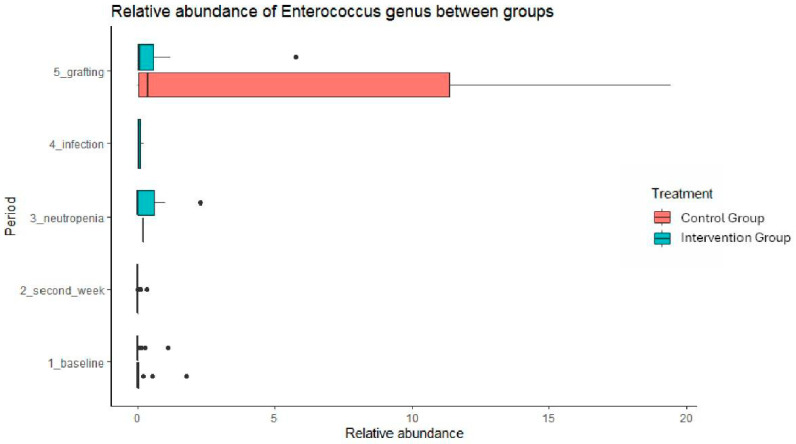
Relative abundance of *Enterococcus* genus in the gut microbiome during hematopoietic stem cell transplantation at Hospital das Clínicas da FMUSP.

**Figure 7 antibiotics-13-01010-f007:**
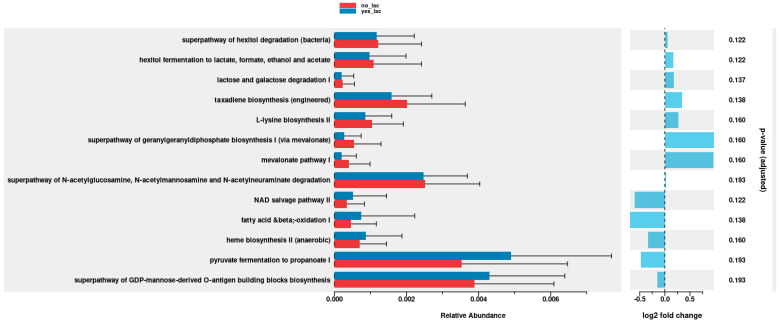
Comparison of metagenome functions of IG and CG using the PICRUSt2 program of gut microbiome of hematopoietic stem cell transplantation patients.

**Table 1 antibiotics-13-01010-t001:** Clinical, demographic, and transplant characteristics of HSCT patients comparing intervention and control groups.

Characteristics of the Patients	Intervention Group	Group Control	OR (95% CI)	*p*-Value
	N = 22	N = 20		
**Diagnosis**				
Gamopathy	6 (27%)	6 (30%)	0.56 (0.24–5.38)	1
Hodgkin’s lymphoma	3 (14%)	4 (20%)	1.56 (0.22–12.3)	0.691
Non-Hodgkin’s lymphoma	7 (32%)	5 (25%)	0.72 (0.14–3.47)	0.739
Acute lymphocytic leukemia	1 (05%)	2 (10%)	0.0 (0.0–42.9)	0.608
Acute myeloid leukemia	4 (18%)	2 (10%)	0.50 (0.04–21.20)	0.598
Chronic myeloid leukemia	1 (05%)	0		1
Age at HSCT ^1^, yr, median (range)	51 (22–67)	40 (22–70)		0.094
Male sex	10 (46%)	12 (60%)	0.56 (0.13–2.22)	0.527
**Colonization**				
Vancomycin-resistant *Enterococcus*	8 (36%)	11 (55%)	2.09 (0.52–8.81)	0.352
Vancomycin-resistant *Enterococcus* + *Acinetobacter baumanni*	0	1 (05%)		0.488
Vancomycin-resistant *Enterococcus* + *Klebsiella pneumonia*	5 (23%)	5 (25%)	1.12 (0.21–5.94)	1
Vancomycin-resistant *enterococcus* + *Klebsiella pneumoniae + Acinetobacter baumanni*	4 (18%)	1 (5%)	0.24 (0.004–2.78)	0.347
Carbapenem-resistant *Klebsiella pneumoniae*	2 (09%)	1 (5%)	0.53 (0.008–11.05)	1
Carbapenem-resistant *Klebsiella pneumoniae* + Carbapenem-resistant *E. coli*	1 (05%)	0		1
*E. coli* carbapenem-resistant + *Klebsiella pneumoniae*	1 (05%)	0		1
*Pseudomonas* + *Klebsiella pneumoniae* resistant to carbapenem	0	1 (05%)		0.488
*Acinetobacter baumanni*	1 (05%)	0		1
**HSCT** ^1^				
Autologous	16 (73%)	16 (80%)		0.849
Allogeneic	4 (18%)	4 (20%)	1 (0.15–6.3)	1
Did not undergo HSCT	2 (09%)	0		
**HSCT** ^1^ **Conditioning Therapy** ^2^				
Myeloablative	18 (90%)	19 (95%)		1
Reduced intensity	2 (10%)	1 (5%)		1
**Length of hospitalization, days, average (range)** ^2^	**32.2 (±19)**	**26.8 (±14)**		**0.762**
Autologous	30.6 (±21)	22.4 (±8)		
Allogeneic	38.3 (±6)	44.3 (±23)		
**Time until grafting, days, average (range)** ^2^	**9 (4–21)**	**9 (6–14)**		**0.667**
**Toxicities in HSCT** ^3^				
Diarrhea	15 (75%)	19 (95%)	8.46 (0.92–419)	0.815
Severe diarrhea (grades III and IV)	5 (25%)	1 (5%)	0.18 (0.0036–1.98)	0.193
Mucositis	13 (65%)	17 (85%)	3.79 (0.74–26.10)	0.808
Severe mucositis (grades III and IV)	8 (40%)	7 (35%)	1.16 (0.28–4.84)	1.000
**Nutritional status**				**0.594**
Thinness	0	1 (5%)		
Normal	8 (36%)	6 (30%)		
Overweight	7 (32%)	7 (35%)		
Obese	7 (32%)	6 (30%)		

^1^ Hematopoietic stem cell transplantation (HSCT). ^2^ We considered only patients undergoing HSCT. ^3^ Gastrointestinal toxicities (mucositis and diarrhea) were measured by Common Terminology Criteria for Adverse Events (CTCAE—Version 5.0).

**Table 2 antibiotics-13-01010-t002:** Details of bloodstream infections in patients undergoing hematopoietic stem cell transplantation.

	Intervention Group (n = 19)	Control Group (n = 20)	OR 95%CI	*p*-Value
Number of infections	7 (35%)	4 (20%)		0.3008
**BSI ^1^**	0	2 (10%)		0.4872
Days TCTH to BSI (mean)	5 (±3)	8 (±3)		
BSI during neutropenia	5	4		
BSI during use of *L. plantarum*	2 *	-		
Days free *L. plantarum* (mean)	1(±2)	-		
**Agents**				
**CLABSI ^2^**				
*Capnocytophaga* sp.	0	1 (5%)		
*Staphylococcus epidermidis*	0	1 (5%)		
**MBI-LCBI ^3^**	7 (35%)	2 (10%)	0.21 (0.018–1.37)	0.0648
*Klebsiella pneumonia* MDRO	1 (5%)	0		
*Klebsiella pneumonia*	2 (10%)	0		
*Enterobacter cloacae* complex and *Klebsiella pneumonia*	1 (5%)	0		
*Klebsiella pneumonia* and *Streptococcus oralis*	1 (5%)	0		
*Escherichia coli*	2 (10%)	1 (5%)		
*Escherichia coli* and *Streptococcus viridans*	0	1 (5%)		
Same agent as previous colonization #	3 (15%)	0		

^1^ Bloodstream infection (BSI); ^2^ central line-associated bloodstream infection (CLABSI); ^3^ mucosal barrier injury–laboratory-confirmed bloodstream infection (MBI-LCBI); * both *Klebsiella pneumoniae*. # Carbapenem-resistant *K. pneumoniae*.

## Data Availability

Microbiome sequencing data are available on Bioproject (NCBI—National Center for Biotechnology Information, U.S. National Library of Medicine) under the accession number PRJNA724885.

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
