# Peer review of "Impact of Exogenous Lactiplantibacillus plantarum on the Gut Microbiome of Hematopoietic Stem Cell Transplantation Patients Colonized by Multidrug-Resistant Bacteria: An Observational Study"

_antibiotics, 2024, doi:10.3390/antibiotics13111010_

Round 1
Reviewer 1 Report (New Reviewer)
Comments and Suggestions for Authors
In this study, they assessed the use of L. plantarum as a probiotic in patients undergoing HSCT who were colonized by MDROs. Overall, this study provides valuable insights into the use of probiotics in immunocompromised patients and gut microbiome modulation in the context of MDRO colonization. I recommend accepting this paper with minor revisions.
Questions:
1. The conclusion is that L. plantarum is safe in HSCT patients colonized by MDROs. However, there was an increased infection ratio observed in the IG group. Can the authors discuss a little bit more about the consequences of these infections?
2. The conclusion of this study is still not clear. Do the results support the hypothesis that L. plantarum inhibits the growth of MDROs? Or should more participants be recruited to further explore the pilot findings in this study?
Author Response
Review 1
In this study, they assessed the use of L. plantarum as a probiotic in patients undergoing HSCT who were colonized by MDROs. Overall, this study provides valuable insights into the use of probiotics in immunocompromised patients and gut microbiome modulation in the context of MDRO colonization. I recommend accepting this paper with minor revisions.
Questions:
- The conclusion is that L. plantarum is safe in HSCT patients colonized by MDROs. However, there was an increased infection ratio observed in the IG group. Can the authors discuss a little bit more about the consequences of these infections?
In the results there is this paragraph : “ The intervention group developed BSI in the period of neutropenia after L . plantarum was discontinued. We observed an intestinal domination by Enterobacterales in only one patient who developed BSI.”
In the discussion section we provided more detail about our findings: “ While a significant reduction in MDRO colonization or infection following L. plantarum ingestion compared to the control group was not observed, all infections in the IG were MBI-LCBI that occurring after the cessation of L. plantarum use. Despite individuals in the IG being more frequently colonized by three MDROs (VRE, carbapenem-resistant K. pneumoniae, and carbapenem-resistant A. baumannii.), polymicrobial MDRO-BSI and carbapenem-resistant A. baumannii -BSI were not identified in the intervention group, suggesting potential benefits of L. plantarum use.
- The conclusion of this study is still not clear. Do the results support the hypothesis that L. plantarum inhibits the growth of MDROs? Or should more participants be recruited to further explore the pilot findings in this study?
We agree that more studies with a larger number of individuals are necessary; thus, we highlighted that further studies with more participants are necessary to confirm our findings.
The conclusion: “In conclusion, the ingestion of L. plantarum capsules was deemed safe and well-tolerated. While there may be an impact on Enterococcus abundance and a reduction in Turicibacter, no significant impact on Gram-Negative MDRO abundance in HSCT patients was observed. Further studies with more participants are necessary to confirm our findings.
We added the following phrase to the conclusion; “ Further studies with more participants are necessary to confirm our findings”
Reviewer 2 Report (New Reviewer)
Comments and Suggestions for Authors
The authors of the manuscript "Impact of exogenous Lactiplantibacillus plantarum on the gut microbiome of hematopoietic stem cell transplantation patients colonized by multi-drug resistant bacteria" has found the change in overall microbiome diversity, with the reduction of some level of MDROs, though not significant with the conclusion mentioning the safe use of L. plantarum as probiotics in HSCT patients. The paper is well written, and the methodology well described. Though the study does not provide any striking findings and conclusion, it can be a source of further study in the field.
Author Response
Comments and Suggestions for Authors
The authors of the manuscript "Impact of exogenous Lactiplantibacillus plantarum on the gut microbiome of hematopoietic stem cell transplantation patients colonized by multi-drug resistant bacteria" has found the change in overall microbiome diversity, with the reduction of some level of MDROs, though not significant with the conclusion mentioning the safe use of L. plantarum as probiotics in HSCT patients. The paper is well written, and the methodology well described. Though the study does not provide any striking findings and conclusion, it can be a source of further study in the field.
We agree and added to the conclusion the following phrase ;
We agree that studies with a larger number of individuals are necessary; thus, we highlighted that further studies with more participants are necessary to confirm our findings.
The conclusion: “In conclusion, the ingestion of L. plantarum capsules was deemed safe and well-tolerated. While there may be an impact on Enterococcus abundance and a reduction in Turicibacter, no significant impact on Gram-Negative MDRO abundance in HSCT patients was observed. Further studies with more participants are necessary to confirm our findings.
Reviewer 3 Report (New Reviewer)
Comments and Suggestions for Authors
The study describes an intervention study with a known probiotic, L. plantarum in HSCT patients, with the aim to reduce the frequent complication of BSI (or other infections). The study is interesting and novel - as the authors point out, there is some work published with pediatric patients only. This line of research is certainly promising and therefore the study is appropriate and the results worthy.
However the text needs substancial improvement. The Figures are poorly presented and need to be improved. The description of the results obtained is very difficult to follow. Some examples that need clarification:
- Figure 2: please indicate which results belong to IG and CG.
- Figure 4: are these results only for the IG patients before thier BSI? Please identify these results at the figure caption
- Figure 4: IG or CG???
- Figure 7 is inneligible and the caption needs more detail
- line 207: "The Lactobacillaceae family was also identified as differentially abundant when the alpha value for the factorial Kruskal-Wallis test was p<0.01 and the LDA >3" Between what?? At what time-point???
- line 265: "Most patients enrolled in our study underwent autologous HSCT". Please provide details about this.
- how was safety and tolerability of the capsules demonstrated? This has to be more clarified in the text
K. pneumoniae infections in patients in the IG were alarmingly high, and the finding that IG developed MBI-262 LCBI during the neutropenia period, after L. plantarum was discontinued is a disconcerting finding. Please discuss the implications of this negative result.
- line 347: how were L. plantarum capsules formulated? At lest the list of excipients should be provided
Author Response
The study describes an intervention study with a known probiotic, L. plantarum in HSCT patients, with the aim to reduce the frequent complication of BSI (or other infections). The study is interesting and novel - as the authors point out, there is some work published with pediatric patients only. This line of research is certainly promising and therefore the study is appropriate and the results worthy.
However the text needs substancial improvement. The Figures are poorly presented and need to be improved. The description of the results obtained is very difficult to follow. Some examples that need clarification:
- Figure 2: please indicate which results belong to IG and CG.
We divided the figure 2 in A (IG and CG ) and B (IG) .
We updated the legend : Figure 2. Alteration in Alpha diversity by Observed ASV, Shannon, Simpson and Pielou’s evenness index among HSCT patients between periods for both IG and CG (A- IG and CG and B IG). The alpha diversity indexes were tested for differences using Wilcoxon rank sum test (*p<0.05 ;**p<0.01). The continuous line represents the median and the shaded region the 25 and 75 percentiles.
- Figure 4: are these results only for the IG patients before thier BSI? Please identify these results at the figure caption
The figure 4 represents all the individuals belong to the Intervention Group (IG) included the individuals that developed BSI.
- Figure 4: IG or CG???
Figure 4 included all the individuals in the IG.
- Figure 7 is inneligible and the caption needs more detail.
Sorry about this, we modified figure 7 and updated it as suggested.
The PICRUSt2 program was used to predict the metagenome functions of the gut microbiome of HSCT patients. We observed a greater, though statistically non-significant, relative abundance of pyruvate fermentation to propanoate I (p=0.193) in the IG (yes-lac) compared to the CG (no-Lac) (Fig 7).
- line 207: "The Lactobacillaceae family was also identified as differentially abundant when the alpha value for the factorial Kruskal-Wallis test was p<0.01 and the LDA >3" Between what?? At what time-point???
After two weeks of use.
We highlighted this information.
- line 265: "Most patients enrolled in our study underwent autologous HSCT". Please provide details about this.
The details are on table 1. We added a phrase about the demographic and clinical data of enrolled patients.
Thirty-two patients enrolled in the study were autologous HSCT, most of the patients received myeloablative treatment and Lymphoma and Gammopathy were the most frequent underlying diseases.
- how was safety and tolerability of the capsules demonstrated? This has to be more clarified in the text
We added the following paragraphs in methods, results and discussion.
Methods: All patients were instructed about the use of L. plantarum capsules. The product was dispensed by the unit's pharmacy with subsequent analysis of the amount consumed to assess adherence to the proposed therapy. The feasibility of using the probiotic was defined by consumption of 50% of the recommended dose by 75% of the patients in the study. The safety of the use of probiotics was due to the development of BSI caused by L. plantarum from the beginning of use until D+60. Any symptoms caused by the probiotic were monitored daily.
Results: “The only symptom reported by the individuals was flatulence (mild flatulence (grade 1 by CTCAE) or moderate flatulence (grade 2) that were reported by three individuals.”
- pneumoniae infections in patients in the IG were alarmingly high, and the finding that IG developed MBI-262 LCBI during the neutropenia period, after L. plantarum was discontinued is a disconcerting finding. Please discuss the implications of this negative result.
There is this paragraph in the discussion section: “While a significant reduction in MDRO colonization or infection following L. plantarum ingestion compared to the control group was not observed, all infections in the IG were MBI-LCBI that occurring after the cessation of L. plantarum use. Despite individuals in the IG being more frequently colonized by three MDROs (VRE, carbapenem-resistant K. pneumoniae, and carbapenem-resistant A. baumannii.), polymicrobial MDRO-BSI and carbapenem-resistant A. baumannii -BSI were not identified in the intervention group, suggesting potential benefits of L. plantarum use”.
- line 347: how were L. plantarum capsules formulated? At lest the list of excipients should be provided.
The following information is in the methods section:
- plantarum, G18 lineage, was administrated as a probiotic to the treatment group at a dose of 5 x 109 Colony Forming Units (CFU) twice daily in the form of gastric release capsules produced in a compounding pharmacy. The HSCT pharmacy supervised capsule dispensing and assessed drug-target engagement quantification. Consumption was terminated in cases of HSCT neutropenia. All symptoms in those taking L. plantarum were assessed according to CTCAE protocol [30]. The safety of the use of L. plantarum was also evaluated by observing the development of BSI by L. plantarum up to day +60 after its use. The content of L. plantarum capsules was analyzed by 16S rRNA next-generation sequencing (NGS), growth in Lactiplantibacillus culture medium (MRS Agar), and mass spectrometry (Maldi-TOF-Buker-Germany).
The study describes an intervention study with a known probiotic, L. plantarum in HSCT patients, with the aim to reduce the frequent complication of BSI (or other infections). The study is interesting and novel - as the authors point out, there is some work published with pediatric patients only. This line of research is certainly promising and therefore the study is appropriate and the results worthy.However the text needs substancial improvement. The Figures are poorly presented and need to be improved. The description of the results obtained is very difficult to follow. Some examples that need clarification:
- Figure 2: please indicate which results belong to IG and CG.
We divided the figure 2 in A (IG and CG ) and B (IG) .
We updated the legend : Figure 2. Alteration in Alpha diversity by Observed ASV, Shannon, Simpson and Pielou’s evenness index among HSCT patients between periods for both IG and CG (A- IG and CG and B IG). The alpha diversity indexes were tested for differences using Wilcoxon rank sum test (*p<0.05 ;**p<0.01). The continuous line represents the median and the shaded region the 25 and 75 percentiles.
- Figure 4: are these results only for the IG patients before thier BSI? Please identify these results at the figure caption
The figure 4 represents all the individuals belong to the Intervention Group (IG) included the individuals that developed BSI.
- Figure 4: IG or CG???
Figure 4 included all the individuals in the IG.
- Figure 7 is inneligible and the caption needs more detail.
Sorry about this, we modified figure 7 and updated it as suggested.
The PICRUSt2 program was used to predict the metagenome functions of the gut microbiome of HSCT patients. We observed a greater, though statistically non-significant, relative abundance of pyruvate fermentation to propanoate I (p=0.193) in the IG (yes-lac) compared to the CG (no-Lac) (Fig 7).
- line 207: "The Lactobacillaceae family was also identified as differentially abundant when the alpha value for the factorial Kruskal-Wallis test was p<0.01 and the LDA >3" Between what?? At what time-point???
After two weeks of use.
We highlighted this information.
- line 265: "Most patients enrolled in our study underwent autologous HSCT". Please provide details about this.
The details are on table 1. We added a phrase about the demographic and clinical data of enrolled patients.
Thirty-two patients enrolled in the study were autologous HSCT, most of the patients received myeloablative treatment and Lymphoma and Gammopathy were the most frequent underlying diseases.
- how was safety and tolerability of the capsules demonstrated? This has to be more clarified in the text
We added the following paragraphs in methods, results and discussion.
Methods: All patients were instructed about the use of L. plantarum capsules. The product was dispensed by the unit's pharmacy with subsequent analysis of the amount consumed to assess adherence to the proposed therapy. The feasibility of using the probiotic was defined by consumption of 50% of the recommended dose by 75% of the patients in the study. The safety of the use of probiotics was due to the development of BSI caused by L. plantarum from the beginning of use until D+60. Any symptoms caused by the probiotic were monitored daily.
Results: “The only symptom reported by the individuals was flatulence (mild flatulence (grade 1 by CTCAE) or moderate flatulence (grade 2) that were reported by three individuals.”
- pneumoniae infections in patients in the IG were alarmingly high, and the finding that IG developed MBI-262 LCBI during the neutropenia period, after L. plantarum was discontinued is a disconcerting finding. Please discuss the implications of this negative result.
There is this paragraph in the discussion section: “While a significant reduction in MDRO colonization or infection following L. plantarum ingestion compared to the control group was not observed, all infections in the IG were MBI-LCBI that occurring after the cessation of L. plantarum use. Despite individuals in the IG being more frequently colonized by three MDROs (VRE, carbapenem-resistant K. pneumoniae, and carbapenem-resistant A. baumannii.), polymicrobial MDRO-BSI and carbapenem-resistant A. baumannii -BSI were not identified in the intervention group, suggesting potential benefits of L. plantarum use”.
- line 347: how were L. plantarum capsules formulated? At lest the list of excipients should be provided.
The following information is in the methods section:
- plantarum, G18 lineage, was administrated as a probiotic to the treatment group at a dose of 5 x 109 Colony Forming Units (CFU) twice daily in the form of gastric release capsules produced in a compounding pharmacy. The HSCT pharmacy supervised capsule dispensing and assessed drug-target engagement quantification. Consumption was terminated in cases of HSCT neutropenia. All symptoms in those taking L. plantarum were assessed according to CTCAE protocol [30]. The safety of the use of L. plantarum was also evaluated by observing the development of BSI by L. plantarum up to day +60 after its use. The content of L. plantarum capsules was analyzed by 16S rRNA next-generation sequencing (NGS), growth in Lactiplantibacillus culture medium (MRS Agar), and mass spectrometry (Maldi-TOF-Buker-Germany).

Reviewer 4 Report (New Reviewer)
Comments and Suggestions for Authors
This is an interesting study, in general. The authors aimed to assess the possible beneficial effects of L. plantarum in patients undergoing hematopoietic stem cell transplantation [HSCT] and were colonized with multidrug-resistant organisms [MDROs].
Generally, they found L. plantarum to be safe, and this is a very helpful information.
However, probiotic has not shown to express an impact on gram [-] MDRO abundance in MDRO gram [-] colonized HSCT patients.
To my opinion this study should have a control group, that is a group receiving probiotic as a prevention measure and not as treatment after MDRO colonization.
I understand that HSCT patients is a very difficult to handle group, but 20 patients in the intervention group, subjected to either autologous [n=16] or allogenic [n=4] transplantation is a rather small number to give reliable results.
Investigation group received probiotic until the onset of neutropenia. Please present data on how many days they have received probiotic; the same for the controls
What was the control group? - do control patients receive a placebo treatment or nothing at all?
Author Response
This is an interesting study, in general. The authors aimed to assess the possible beneficial effects of L. plantarum in patients undergoing hematopoietic stem cell transplantation [HSCT] and were colonized with multidrug-resistant organisms [MDROs].
Generally, they found L. plantarum to be safe, and this is a very helpful information.
However, probiotic has not shown to express an impact on gram [-] MDRO abundance in MDRO gram [-] colonized HSCT patients.
Yes, we agree. However as described in results and in the discussion section: “There is a decreased on VRE abundance on the intervention group with a trend to be significant. We highlighted this data on the results , discussion and conclusion”.
To my opinion this study should have a control group, that is a group receiving probiotic as a prevention measure and not as treatment after MDRO colonization.
I understand that HSCT patients is a very difficult to handle group, but 20 patients in the intervention group, subjected to either autologous [n=16] or allogenic [n=4] transplantation is a rather small number to give reliable results.
We agree that the size of population was small and that a control group that received placebo would be a more elegant study.
Participants were allocated to an intervention group (n=22) that received L. plantarum capsules (5x109 CFU) twice daily before HSCT until the onset of neutropenia and a control group (n=20).
These questions were addressed in the limitation paragraph.
Investigation group received probiotic until the onset of neutropenia. Please present data on how many days they have received probiotic; the same for the controls.
The mean duration of L. plantarum use was 19 days.
What was the control group? - do control patients receive a placebo treatment or nothing at all?
The control did not receive placebo.
A paragraph of limitations in the discussion pointed out all these problems.
This study has limitations. For example, the duration of L. plantarum consumption varied among patients based on their availability and clinical condition for admission to HSCT. Additionally, the sample size was too small and not controlled for clinical features that could potentially influence the results.
This is an interesting study, in general. The authors aimed to assess the possible beneficial effects of L. plantarum in patients undergoing hematopoietic stem cell transplantation [HSCT] and were colonized with multidrug-resistant organisms [MDROs].
Generally, they found L. plantarum to be safe, and this is a very helpful information.
However, probiotic has not shown to express an impact on gram [-] MDRO abundance in MDRO gram [-] colonized HSCT patients.
Yes, we agree. However as described in results and in the discussion section: “There is a decreased on VRE abundance on the intervention group with a trend to be significant. We highlighted this data on the results , discussion and conclusion”.
To my opinion this study should have a control group, that is a group receiving probiotic as a prevention measure and not as treatment after MDRO colonization.
I understand that HSCT patients is a very difficult to handle group, but 20 patients in the intervention group, subjected to either autologous [n=16] or allogenic [n=4] transplantation is a rather small number to give reliable results.
We agree that the size of population was small and that a control group that received placebo would be a more elegant study.
Participants were allocated to an intervention group (n=22) that received L. plantarum capsules (5x109 CFU) twice daily before HSCT until the onset of neutropenia and a control group (n=20).
These questions were addressed in the limitation paragraph.
Investigation group received probiotic until the onset of neutropenia. Please present data on how many days they have received probiotic; the same for the controls.
The mean duration of L. plantarum use was 19 days.
What was the control group? - do control patients receive a placebo treatment or nothing at all?
The control did not receive placebo.
A paragraph of limitations in the discussion pointed out all these problems.
This study has limitations. For example, the duration of L. plantarum consumption varied among patients based on their availability and clinical condition for admission to HSCT. Additionally, the sample size was too small and not controlled for clinical features that could potentially influence the results.
Round 2
Reviewer 4 Report (New Reviewer)
Comments and Suggestions for Authors
I consider that your answers regarding control group do not cover my expectations
Line 125: The mean duration of L. plantarum use was 19 days . This sentence should be completes with SD or range or IQR, which is very informative
Lines 313 -315: Limitations should be presented in detail and not "for example". Additionally, another major limitation is that this study is, practically, an observational study, since the control group have received nothing at all, and no placebo treatment. This should clearly be mentioned in limitations, and I suggest to be prominent from the title
Line 114: not Lactobacillus - use the new nomenclature throughout the text
Line 356 " L. plantarum consumption was terminated in cases of HSCT neutropenia." Although a placebo group don't exist, I would like to know the mean time of neutropenia induction in the non-treated patients
Author Response
I consider that your answers regarding control group do not cover my expectations
Line 125: The mean duration of L. plantarum use was 19 days . This sentence should be completes with SD or range or IQR, which is very informative
We very sorry that was not clear that we were describing only the individuals that developed BSI. We added the information regarding the entire IG.
We added the following information:
The mean duration of L. plantarum use was 48 days (ranged from 13 to 108 days) in the IG; among individuals that developed BSI was 19 days (ranged from 13-47 days).
Lines 313 -315: Limitations should be presented in detail and not "for example". Additionally, another major limitation is that this study is, practically, an observational study, since the control group have received nothing at all, and no placebo treatment. This should clearly be mentioned in limitations, and I suggest to be prominent from the title
Thank you for the suggestion.
The title and limitation now include the type of study as recomended.
We replaced the tittle as suggested by “ Impact of exogenous Lactiplantibacillus plantarum on the gut microbiome of hematopoietic stem cell transplantation patients colonized by multi-drug resistant bacteria: an observational study”.
We added the information that it was an observational study in the abstract as well.
We added the following paragraph to the discussion section:
This study has several limitations it is an observational study without a control group that received placebo, the duration of L. plantarum consumption varied among patients based on their availability and clinical condition for admission to HSCT. Additionally, the sample size was small and not controlled for clinical features that could potentially influence the results.
Line 114: not Lactobacillus - use the new nomenclature throughout the text
We corrected the nomenclature as recomended .
Line 356 " L. plantarum consumption was terminated in cases of HSCT neutropenia." Although a placebo group don't exist, I would like to know the mean time of neutropenia induction in the non-treated patients
The mean duration of neutropenia was 13 days (10-20 days) in the IG and 13 days (10-18 days) in CG.
We added this information in the results section.
This manuscript is a resubmission of an earlier submission. The following is a list of the peer review reports and author responses from that submission.
Round 1
Reviewer 1 Report
Comments and Suggestions for Authors
In Article “Impact of exogenous Lactobacillus plantarum on the gut microbiome of hematopoietic stem cell transplantation patients colonized by multi-drug resistant bacteria.” reported the study of the impact of Lactobacillus plantarum on modulation of the gut icrobiome in HSCT patients colonized by MDROs. The Phylogenetic Investigation of Communities by Reconstruction of Unobserved States) (PICRUSt2) program was used for prediction of metagenome functions. L. plantarum had an average 86% (±11%) drug-target engagement at 43(±29) days of consumption and it was safe, tolerable, and associated with an increase in the abundance of the Lactobacillales (p=0.004). The authors also study the greater non-significant pyruvate fermentation to propanoate I (p=0.193) relative abundance in the IG comparing with CG. L. plantarum use was safe and tolerable by HSCT patients and it has no effect on Gram Negative MDRO abundance in MDRO gram-negative colonized HSCT patients.
The following some corrections are recommended:
The Cited Reference is not in the format of journal guidelines. Check all the cited reference year, volume and page number and keep to a journal guideline.
Author Response
Reviewer 1
In Article “Impact of exogenous Lactobacillus plantarum on the gut microbiome of hematopoietic stem cell transplantation patients colonized by multi-drug resistant bacteria.” reported the study of the impact of Lactobacillus plantarum on modulation of the gut icrobiome in HSCT patients colonized by MDROs. The Phylogenetic Investigation of Communities by Reconstruction of Unobserved States) (PICRUSt2) program was used for prediction of metagenome functions. L. plantarum had an average 86% (±11%) drug-target engagement at 43(±29) days of consumption and it was safe, tolerable, and associated with an increase in the abundance of the Lactobacillales (p=0.004). The authors also study the greater non-significant pyruvate fermentation to propanoate I (p=0.193) relative abundance in the IG comparing with CG. L. plantarum use was safe and tolerable by HSCT patients and it has no effect on Gram Negative MDRO abundance in MDRO gram-negative colonized HSCT patients.
The following some corrections are recommended:
The Cited Reference is not in the format of journal guidelines. Check all the cited reference year, volume and page number and keep to a journal guideline.
We fixed the references, and they are now in the format of the journal.

Reviewer 2 Report
Comments and Suggestions for Authors
The article explores the potential of multidrug-resistant bacteria (MDROs) in causing early bloodstream infections in patients undergoing hematopoietic stem cell transplantation (HSCT). It investigates the impact of externally administered Lactobacillus plantarum on the colonization of MDROs in the intestines of HSCT patients.
Specific recommendations:
1、Abstract: The description is disorganized. Please structure the abstract by separately describing the research background, methods, results, and conclusions.
2、Abstract: Add keywords.
3、Introduction: The last sentence of the first paragraph about probiotics would be more appropriate at the beginning of the second paragraph.
4、Provide an in-depth introduction to the definition and biological functions of probiotics. Additionally, elaborate on the current mechanisms by which probiotics are used to improve inflammation.
5、The connection between intestinal MDROs and early infection events in HSCT patients is established. Lactobacillus plantarum, as a probiotic, acts through the gut microbiota, but there is no mention of gut microbiota in the introduction.
6、The "Results" section lacks logical coherence. The patient selection process in the "Patients Characteristic" part belongs to the methods and should be described there. Figure 1 should be placed in the "Methods—Study Design" section, and exclusion criteria should be supplemented in Figure 1. Where is Table 1? Tables 2 and 3 need to include statistical values.
7、In multiple quantitative analysis result graphs, the bars exceed the chart range. Please provide an overview result chart. Additionally, provide the Z/Chi-square values for statistical analysis results.
Author Response
Reviewer 2
The article explores the potential of multidrug-resistant bacteria (MDROs) in causing early bloodstream infections in patients undergoing hematopoietic stem cell transplantation (HSCT). It investigates the impact of externally administered Lactobacillus plantarum on the colonization of MDROs in the intestines of HSCT patients.
Specific recommendations:
1、Abstract: The description is disorganized. Please structure the abstract by separately describing the research background, methods, results, and conclusions.
We structured the abstract as recommended.
2、Abstract: Add keywords.
We added the followed keywords.
Lactiplantibacillus plantarum ; gut microbiome, Gut colonization by multidrug-resistant organisms; Hematopoietic Stem Cell Transplantation.
3、Introduction: The last sentence of the first paragraph about probiotics would be more appropriate at the beginning of the second paragraph.
We moved the last sentence of the first paragraph about probiotics as recommended to the beginning of the second paragraph.
4、Provide an in-depth introduction to the definition and biological functions of probiotics. Additionally, elaborate on the current mechanisms by which probiotics are used to improve inflammation.
We added the following paragraph about probiotics and add information about mechanism by which probiotics are used to improve inflammation.
Few studies, however, have attempted to modulate the presence of MDROs in the intestinal microbiome by focusing on the individual’s diet or the use of prebiotics, probiotics, and symbiotics [8-11]. Moreover, this has rarely been reported in patients undergoing HSCT. Probiotics are live microorganisms that can provide benefit to the host if present in adequate doses [12]. Usually, probiotics are not offered to immunocompromised patients, like those undergoing HSCT, due to a risk of translocation and infection. To date, only one study demonstrated the safety and feasibility of using a specific probiotic, Lactiplantibacillus plantarum, in pediatric neutropenic patients undergoing HSCT [13]. The presence of Lactiplantibacillus spp. in the fecal microbiota of hospitalized subjects has been associated with a lack of MDRO acquisition, indicating a potential protective role for these bacteria in HSCT [14]. Lactiplantibacillus plantarum can have modulatory effects on the intestine in healthy individuals by reducing proteins such as matrix metallopeptidases and zonulin [15]. However, a recent study described a cluster, using whole-genome sequencing, of BSI due to Lactiplantibacillus in pediatric patients undergoing HSCT that received probiotics [16]. Thus, more studies on the safety use of probiotics in HSCT are needed.
5、The connection between intestinal MDROs and early infection events in HSCT patients is established. Lactobacillus plantarum, as a probiotic, acts through the gut microbiota, but there is no mention of gut microbiota in the introduction.
We added a paragraph about gut microbiome in the introduction section as suggested.
Gut microbiome diversity has been associated with diet, antibiotic use, and chemotherapy in HSCT patients. Higher diversity of gut microbiome at the time of neutrophil engraftment was associated with lower mortality and graft-versus-host diseases in allogeneic HSCT [4]. The use of anti-anaerobic antibiotics decreased gut microbiome diversity in allogeneic HSCT patients [5]. Moreover, the impact of using antibiotics with different anaerobic spectra of activity and fluoroquinolone prophylaxis on gut microbiome is controversial. A recent study showed no significant impact of ending ciprofloxacin prophylaxis on microbiome diversity [6].
6、The "Results" section lacks logical coherence. The patient selection process in the "Patients Characteristic" part belongs to the methods and should be described there. Figure 1 should be placed in the "Methods—Study Design" section, and exclusion criteria should be supplemented in Figure 1. Where is Table 1? Tables 2 and 3 need to include statistical values.
We added the information regarding the exclusion criteria as legend of figure 1 and we moved the figure 1 to methods section as recommended.
Exclusion criteria: “Patients who did not collect stool samples after initiation of HSCT conditioning”.
We fixed the number of tables. Table 1 was excluded as the reference were cited in the text [27-30]and its described only the primers used in the methodology.
We included the OR and 95% CI on Table 2 and table 3 now named table 1 and table 2 .
Table 1-. Clinical, demographic, and transplant characteristics of HSCT patients comparing intervention and control group.
|
Characteristics of the patients |
Intervention Group |
Group control |
OR (95% CI) |
P value |
|
|
N=22 |
N=20 |
|
|
|
Diagnosis |
|
|
|
|
|
Gamopathy |
6 (27%) |
6 (30%) |
0.56(0.24-5.38)
|
1 |
|
Hodgkin's lymphoma |
3 (14%) |
4 (20%) |
1,56 (0.22-12,3) |
0.691 |
|
Non-Hodgkin's lymphoma |
7 (32%) |
5 (25%) |
0.72 (0.14-3.47) |
0.739 |
|
Acute lymphocytic leucemia |
1 (05%) |
2 (10%) |
0,0 (0.0-42.9) |
0.608 |
|
Acute myeloid leucemia |
4 (18%) |
2 (10%) |
0.50 (0.04-21.20) |
0.598 |
|
Chronic myeloid leucemia |
1 (05%) |
0 |
|
1 |
|
Age at HSCT1, yr, median (range) |
51 (22-67) |
40 (22-70) |
|
0.094 |
|
Male sex |
10 (46%) |
12 (60%) |
0.56(0.13-2.22) |
0.527 |
|
Colonization |
|
|
|
|
|
Vancomycin resistant enterococcus |
8 (36%) |
11 (55%) |
2.09(0.52-8.81) |
0.352 |
|
Vancomycin-resistant Enterococcus + Acinetobacter baumanni |
0 |
1(05%) |
|
0.488 |
|
Vancomycin-resistant Enterococcus + Klebsiella pneumoniae |
5 (23%) |
5 (25%) |
1.12(0.21-5.94) |
1 |
|
Vancomycin-resistant enterococcus + Klebsiella pneumoniae + Acinetobacter baumanni |
4 (18%) |
1 (5%) |
0.24(0.004-2.78) |
0.347 |
|
Carbapenem-resistant Klebsiella pneumoniae |
2 (09%) |
1 (5%) |
0.53(0.008-11.05) |
1 |
|
Carbapenem-resistant Klebsiella pneumoniae + Carbapenem-resistant E. coli |
1 (05%) |
0 |
|
1 |
|
E. coli carbapenem-resistant + Klebsiella pneumoniae |
1 (05%) |
0 |
|
1 |
|
Pseudomonas + Klebsiella pneumoniae resistant to carbapenem |
0 |
1(05%) |
|
0.488 |
|
Acinetobacter baumanni |
1 (05%) |
0 |
|
1 |
|
HSCT1 |
|
|
|
|
|
Autologous |
16 (73%) |
16 (80%) |
|
0.849 |
|
Allogeneic |
4 (18%) |
4 (20%) |
1 (0.15-6.3) |
1 |
|
Did not undergoing HSCT |
2 (09%) |
0 |
|
|
|
HSCT1 Conditioning Therapy 2 |
|
|
|
|
|
Myeloablative |
18 (90%) |
19 (95%) |
|
1 |
|
Reduced intensity |
2(10%) |
1 (5%) |
|
1 |
|
Length of hospitalized, days, average (range) 2 |
32.2 (±19) |
26.8 (±14) |
|
0.762 |
|
Autologous |
30.6 (±21) |
22.4 (±8) |
|
|
|
Allogeneic |
38.3 (±6) |
44.3 (±23) |
|
|
|
Time until grafting, days, average (range) 2 |
9 (4-21) |
9 (6-14) |
|
0.667 |
|
Toxicities in HSCT3 |
|
|
|
|
|
Diarrhea |
15 (75%) |
19 (95%) |
8.46(0.92-419) |
0.815 |
|
Severe diarrhea (grades III and IV) |
5 (25%) |
1 (5%) |
0.18 (0.0036-1.98) |
0.193 |
|
Mucositis |
13 (65%) |
17 (85%) |
3.79 (0.74-26.10) |
0.808 |
|
Severe mucositis (grades III and IV) |
8 (40%) |
7 (35%) |
1.16 (0.28-4.84) |
1,000 |
|
Nutritional status |
|
|
|
0.594 |
|
Thinness |
0 |
1 (5%) |
|
|
|
Normal |
8 (36%) |
6 (30%) |
|
|
|
Overweight |
7 (32%) |
7 (35%) |
|
|
|
Obese |
7 (32%) |
6 (30%) |
|
|
7、In multiple quantitative analysis result graphs, the bars exceed the chart range. Please provide an overview result chart. Additionally, provide the Z/Chi-square values for statistical analysis results.
We fixed the bars, and we hope that now the graphs are clear.

Reviewer 3 Report
Comments and Suggestions for Authors
Overall, the entire manuscript is written sloppily and was difficult to read. It is not written in typically scientific language. Incorrect line numbering prevents proper review of the manuscript.
The description of the introduction is quite poor and contains few references, which are also old. A hypothesis is given, but it is not even indicated what the purpose of this work is?
Writing about probiotic bacteria without specifying the exact strain or line is shameful. The species name L. plantarum is incorrect. Please see the new nomenclature for lactic acid bacteria. GenBank number is missing!
The research group is small. Is it possible to draw conclusions since there were so few patients?
Below are some comments on the manuscript. In general, the entire manuscript should be considered for resubmission after revisions.
Please fill in missing spaces or remove double spaces throughout the manuscript.
The manuscript should be written in the passive voice and the statements "We evaluated..." and "We observed..." should be avoided.
Please specify keywords.
Figure 1. Abbreviations should be explained under the figure title.
Where is table 1?
Table 2, line 186. Enterococcus is written in capital letters and in italics.
Line 91. This is not how you write the number of bacteria - 6x108 ±3.45 x108. What is the UFC unit? Since capsules are indicated, the unit should be formulated as CFU/g.
Table 3. Abbreviations should be explained below the table.
Figure 2 and 3. The charts are completely illegible and unsigned. Explanation of abbreviations is missing.
The remaining figures are chaotically pasted between the descriptions. There are no descriptions on the axes, they are illegible images generated in the program.
The discussion does not refer to the results presented in the manuscript.
There are no literature references to the Methods section.
The summary of the results is modest and does not refer to the hypothesis (the aim of the work was not set at all).
The proposed literature is not new. The literature should be replaced and supplemented with the latest available literature.
Author Response
Reviewer 3
Overall, the entire manuscript is written sloppily and was difficult to read. It is not written in typically scientific language. Incorrect line numbering prevents proper review of the manuscript.
The description of the introduction is quite poor and contains few references, which are also old. A hypothesis is given, but it is not even indicated what the purpose of this work is?
We added the following paragraphs to the introduction section:
“Gut microbiome diversity has been associated with diet, antibiotic use, and chemotherapy in HSCT patients. Higher diversity of gut microbiome at the time of neutrophil engraftment was associated with lower mortality and graft-versus-host diseases in allogeneic HSCT [4]. The use of anti-anaerobic antibiotics decreased gut microbiome diversity in allogeneic HSCT patients [5]. Moreover, the impact of using antibiotics with different anaerobic spectra of activity and fluoroquinolone prophylaxis on gut microbiome is controversial. A recent study showed no significant impact of ending ciprofloxacin prophylaxis on microbiome diversity [6].
Previous gut colonization has been associated with barrier BSI in neutropenic HSCT. At our center, Ferreira et al. evaluated 232 HSCT patients and demonstrated an MDRO colonization rate of 30.5%. Bloodstream infection (BSI) occurred in 20% of the MDRO-positive group. The authors observed, by multivariate analysis, that prior gut-colonization by gram-negative MDROs was a risk factor for this outcome [1]. Previous studies also observed that the increase of Enterococcus in the microbiome at the engraftment period is associated with BSI during the HSCT [7]”. Few studies, however, have attempted to modulate the presence of MDROs in the intestinal microbiome by focusing on the individual’s diet or the use of prebiotics, probiotics, and symbiotics [8-11]. The presence of Lactiplantibacillus spp. in the fecal microbiota of hospitalized subjects has been associated with a lack of MDRO acquisition, indicating a potential protective role for these bacteria in HSCT [14]. Lactiplantibacillus plantarum can have modulatory effects on the intestine in healthy individuals by reducing proteins such as matrix metallopeptidases and zonulin [15]. However, a recent study described a cluster, using whole-genome sequencing, of BSI due to Lactiplantibacillus in pediatric patients undergoing HSCT that received probiotics [16]. Thus, more studies on the safety use of probiotics in HSCT are needed.
Although data on the use of L. plantarum in HSCT is scarce. We updated the references and cited references of gut microbiome of HSCT patients and regarding L. plantarum use and inflammatory modulation that had been published from 2020 to 2023.
Daoud-Asfour H, Henig I, et al. Omitting Ciprofloxacin Prophylaxis in Patients Undergoing Allogeneic Hematopoietic Stem Cell Transplantation and Its Impact on Clinical Outcomes and Microbiome Structure..Transplant Cell Ther. 2022 Mar;28(3):168.e1-168.e8. doi: 10.1016/j.jtct.2021.12.012.
Tanaka JS, Young RR, et al.
Anaerobic Antibiotics and the Risk of Graft-versus-Host Disease after Allogeneic Hematopoietic Stem Cell Transplantation. Biol Blood Marrow Transplant. 2020 Nov;26(11):2053-2060. doi: 10.1016/j.bbmt.2020.07.011.
Silva M B, Ponce DM, Dai A et al. Preservation of the fecal microbiome is associated with reduced severity of graft-versus-host disease..Blood. 2022 Dec 1;140(22):2385-2397.
Nam B, Soo A Kim SA et al. Regulatory effects of Lactobacillus plantarum HY7714 on skin health by improving intestinal condition. PLoS One . 2020 Apr 10;15(4):e0231268. doi: 10.1371/journal.pone.0231268. eCollection 2020.
Gilliam CH, de Cardenas JB, et al. Lactobacillus bloodstream infections genetically related to probiotic use in pediatric hematopoietic cell transplant patients. Infect Control Hosp Epidemiol. 2023 Mar;44(3):484-487. doi: 10.1017/ice.2021.515. Epub 2022 Feb 28.
D'Angelo C, Sudakaran S, et al. Perturbation of the gut microbiome and association with outcomes following autologous stem cell transplantation in patients with multiple myeloma. Leuk Lymphoma . 2023 Jan;64(1):87-97. doi: 10.1080/10428194.2022.2131410. Epub 2022 Oct 11.
Writing about probiotic bacteria without specifying the exact strain or line is shameful. The species name L. plantarum is incorrect. Please see the new nomenclature for lactic acid bacteria. GenBank number is missing!
We so sorry about the missed information. We used the L. plantarum G18 lineage.
We fixed the taxonomy in the entire text.
The GenBank access number was described in the “Data Availability Statement”. We added the information as well as to the results section.
Microbiome sequencing data are available on Bioproject (NCBI - National Center for Biotechnology Information, U.S. National Library of Medicine) under the accession number PRJNA724885.
The research group is small. Is it possible to draw conclusions since there were so few patients?
We agree that the number is small, however, it was an exploratory study. There is a paragraph in the discussion section that reinforced the limitation of the study.
“ This study has limitations. For example, the duration of L. plantarum consumption varied among patients based on their availability and clinical condition for admission to HSCT. Additionally, the sample size was small and not controlled for clinical features that could potentially influence the results.”
Below are some comments on the manuscript. In general, the entire manuscript should be considered for resubmission after revisions.
Please fill in missing spaces or remove double spaces throughout the manuscript.
We removed the double spaces throughout the manuscript.
The manuscript should be written in the passive voice and the statements "We evaluated..." and "We observed..." should be avoided.
Thank you for the suggestion. The manuscript was revised by a native English speaker.
Please specify keywords.
Lactiplantibacillus plantarum ; gut microbiome, Gut colonization by multidrug-resistant organisms; Hematopoietic Stem Cell Transplantation
Figure 1. Abbreviations should be explained under the figure title.
Where is table 1?
We excluded table 1 (as its described only the primers used in the study) and we cited the references (27-30).
Table 2, line 186. Enterococcus is written in capital letters and in italics.
We correct it as recommended.
Line 91. This is not how you write the number of bacteria - 6x108 ±3.45 x108. What is the UFC unit? Since capsules are indicated, the unit should be formulated as CFU/g.
We corrected it and replaced by CFU\ g.
Table 3. Abbreviations should be explained below the table.
We explained the abbreviations as recommended.
Figure 2 and 3. The charts are completely illegible and unsigned. Explanation of abbreviations is missing.
The remaining figures are chaotically pasted between the descriptions. There are no descriptions on the axes, they are illegible images generated in the program.
We modified all the figures definition and added the descriptions on the axes.
The discussion does not refer to the results presented in the manuscript.
We modified the discussion and added the following paragraphs:
Of note, L. plantarum demonstrated safety and tolerability before the onset of neutropenia in all individuals undergoing HSCT, and it may potentially reduce fecal levels of the Enterococcus genus in HSCT patients. There was a decrease in the presence of Roseburia and Coprococcus in the gut microbiome during the grafting period in both IG and CG. The consumption of L. plantarum was associated with a significant increase in Lactobacillales and a decrease in Turicibacter in the gut microbiome of HSCT patients during the second week of usage. None of the patients in the intervention group experienced BSI caused by L. plantarum or severe adverse events.
While a significant reduction in MDRO colonization or infection following L. plantarum ingestion compared to the control group was not observed, all infections in the intervention group were MBI-LCBI, with most occurring after the cessation of L. plantarum use. Despite individuals in the IG being more frequently colonized by three MDROs (VRE, carbapenem-resistant K. pneumoniae, and carbapenem-resistant A. baumannii.), polymicrobial MDRO-BSI and carbapenem-resistant A. baumannii -BSI were not identified in the intervention group, suggesting potential benefits of L. plantarum use.
Data on the impact of Lactiplantibacillus on the prevention of MDRO colonization are controversial and dependent on many variables such as species, antibiotic use, and patient population [8]. A recent randomized controlled study involving 120 patients treated with amoxicillin-clavulanate for 10 days, compared the effects with a 30-day treatment with placebo Saccharomyces boulardii and a probiotic mixture containing Saccharomyces boulardii, Lactiplantibacillus acidophilus CFM, Lactiplantibacillus paracasei , Bifidobacterium lactis 4, and Bifidobacterium lactis . It was noted that treatment with the probiotic mixture led to a significant decline in colonization by Pseudomonas (p = 0.041) and AmpC-producing enterobacteria [11]. On the other hand, another prospective, double-blinded, randomized controlled trial found that Lactiplantibacillus rhamnosus did not prevent MDRO colonization among patients receiving broad-spectrum antibiotics [17].
Most patients enrolled in our study underwent autologous HSCT. Few studies have addressed the gut microbiome in autologous HSCT. A recent study observed loss of alpha diversity at the engraftment timepoint driven by decreases in Blautia, Ruminococcus, and Faecalibacterium genera in 33 patients with multiple myeloma undergoing autologous HSCT [18]. We observed a significant reduction in alpha diversity in the gut microbiome especially between baseline and the second week versus neutropenia and grafting. Additionally, we observed intense intra-individual variability in alpha diversity between the time points, as previously shown by Taur et al [7]. Interestingly, there was a reduction in the uniformity of the species observed between baseline and grafting in both groups, slightly smaller in CG than IG. At baseline, there was a predominance of the phyla Bacteroidetes, followed by Firmicutes and Proteobacteria, considered a healthier gut microbiome. On the other hand, there was a decrease in the presence of Roseburia and Coprococcus, short-chain fatty acid producers in the gut microbiome, during the grafting period in both groups, IG and CG, similar to what was described in allogeneic HSCT patients [19].
There are no literature references to the Methods section.
The references cited on methods section [19-45] are below:
CDC - Center for Disease Control and Prevention. Bloodstream Infection Event (Central Line-Associated Bloodstream Infection and Non-central line-associated Bloodstream Infection). Publ on-line http//www.cdc.gov/nhsn/ [Internet]. 2016 ;( January):1–32. Available from: Centers for Disease Control (CDC)/National Healthcare Safety Network (NHSN). Bloodstream Infection Event (Central Line-Associated Bloodstream Infection and Non-central line-associated Bloodstream Infection.. Available at: http://www.cdc.gov/nhsn/PDFs/pscM
Ribeiro RM, de Souza-Basqueira M, de Oliveira LC, Salles FC, Pereira NB, Sabino EC. An alternative storage method for characterization of the intestinal microbiota through next generation sequencing. Rev Inst Med Trop Sao Paulo, 60 (2018), pp.1–11.
Caporaso JG, Lauber CL, Walters WA, Berg-Lyons D, Lozupone CA, Turnbaugh PJ, et al. Global patterns of 16S rRNA diversity at a depth of millions of sequences per sample. Proc Natl Acad Sci U S A. 108 (2011): 4516–4522.
Bolyen E, Rideout JR, Dillon MR, Bokulich NA, Abnet CC, Al-Ghalith GA, et al. Reproducible, interactive, scalable and extensible microbiome data science using QIIME 2. Nat Biotechnol, 37 (2019): 852–857.
Callahan BJ, McMurdie PJ, Rosen MJ, Han AW, Johnson AJA, Holmes SP. DADA2: High-resolution sample inference from Illumina amplicon data. Nat Methods,13 (2016), pp.581–853.
Janssen S, Mcdonald D, Gonzalez A, Navas-molina JA, Jiang L, Xu Z. Phylogenetic Placement of Exact Amplicon Sequences. mSystems. 3 (2018), pp.1–14.
Mirarab S, Nguyen N, Warnow T. SEPP: SATé-enabled phylogenetic placement. In: Pacific Symposium on Biocomputing. 2012. pp. 247–258.
Faith DP. Conservation evaluation and phylogenetic diversity. Biol Conserv. 61(1992),pp. 1–10.
Lozupone CA, Hamady M, Kelley ST, Knight R. Quantitative and Qualitative Diversity Measures Lead to Different Insights into Factors That Structure Microbial Communities, 73 (2007), pp.1576–1585.
Lozupone C, Knight R. UniFrac : a New Phylogenetic Method for Comparing Microbial Communities. 71 (2005): 8228–8235.
McMurdie PJ, Holmes S. Phyloseq: An R Package for Reproducible Interactive Analysis and Graphics of Microbiome Census Data. PLoS One, 8 (2013), p.4.
Bokulich NA, Kaehler BD, Rideout JR, Dillon M, Bolyen E, Knight R, et al. Optimizing taxonomic classification of marker-gene amplicon sequences with QIIME 2 ’ s q2-feature-classifier plugin. 2018, pp. 1–17.
Mandal S, Treuren W Van, White RA, Eggesbø M, Knight R, Peddada SD. Analysis of composition of microbiomes: a novel method for studying microbial composition. 1 (2015), pp. 1–7.
Segata N, Izard J, Waldron L, Gevers D, Miropolsky L, Garrett WS, et al. Metagenomic biomarker discovery and explanation. Genome Biol [Internet]. 2011;12(6):R60. Available from: http://genomebiology.com/2011/11/6/R60
Chen Yang, Jiahao Mai, Xuan Cao, Aaron Burberry, Fabio Cominelli, Liangliang Zhang, ggpicrust2: an R package for PICRUSt2 predicted functional profile analysis and visualization, Bioinformatics, Volume 39, Issue 8, August 2023, btad470, https://doi.org/10.1093/bioinformatics/btad470
The summary of the results is modest and does not refer to the hypothesis (the aim of the work was not set at all).
The objective of the study was not clear. Thus, we revised it.
The actual objective is described below:
“ In the present study, we assessed the use of L. plantarum (intervention) as a probiotic in patients undergoing HSCT who were colonized by MDROs. We hypothesized that this intervention would reduce MDRO colonization and infection compared to an untreated control group. Additionally, we assessed the impact of the intervention on the gut microbiome and the safety and adverse effects of L. plantarum usage in HSCT patients”.
We achieved our objectives that was the safety and of L. plantarum in HSCT patients and described the impact of L. plantarum on gut microbiome as well.
The proposed literature is not new. The literature should be replaced and supplemented with the latest available literature.
Although data on the use of L. plantarum in HSCT is scarce. We updated the references and cited references of gut microbiome of HSCT patients and regarding L. plantarum use and inflammatory modulation that had been published from 2020 to 2023.
We added the following articles that had been published from 2020 to 2023.
Daoud-Asfour H, Henig I, et al. Omitting Ciprofloxacin Prophylaxis in Patients Undergoing Allogeneic Hematopoietic Stem Cell Transplantation and Its Impact on Clinical Outcomes and Microbiome Structure..Transplant Cell Ther. 2022 Mar;28(3):168.e1-168.e8. doi: 10.1016/j.jtct.2021.12.012.
Tanaka JS, Young RR, et al.
Anaerobic Antibiotics and the Risk of Graft-versus-Host Disease after Allogeneic Hematopoietic Stem Cell Transplantation. Biol Blood Marrow Transplant. 2020 Nov;26(11):2053-2060. doi: 10.1016/j.bbmt.2020.07.011.
Silva M B, Ponce DM, Dai A et al. Preservation of the fecal microbiome is associated with reduced severity of graft-versus-host disease..Blood. 2022 Dec 1;140(22):2385-2397.
Nam B, Soo A Kim SA et al. Regulatory effects of Lactobacillus plantarum HY7714 on skin health by improving intestinal condition. PLoS One . 2020 Apr 10;15(4):e0231268. doi: 10.1371/journal.pone.0231268. eCollection 2020.
Gilliam CH, de Cardenas JB, et al. Lactobacillus bloodstream infections genetically related to probiotic use in pediatric hematopoietic cell transplant patients. Infect Control Hosp Epidemiol. 2023 Mar;44(3):484-487. doi: 10.1017/ice.2021.515. Epub 2022 Feb 28.
D'Angelo C, Sudakaran S, et al. Perturbation of the gut microbiome and association with outcomes following autologous stem cell transplantation in patients with multiple myeloma. Leuk Lymphoma . 2023 Jan;64(1):87-97. doi: 10.1080/10428194.2022.2131410. Epub 2022 Oct 11.

Round 2
Reviewer 2 Report
Comments and Suggestions for Authors
The authors have made many satisfactory amendments to the original study, greatly enhancing its readability. However, there are still significant flaws:
-
1、Abstract: The study background mentions that intestinal colonization by MDROs is a risk factor for early bloodstream infections in HSCT, yet the research aim is "to evaluate the impact of Lactobacillus plantarum on the gut microbiome patterns of HSCT patients colonized with MDROs." Why study Lactobacillus plantarum instead of directly investigating MDROs? Why bypass the gut flora to mention Lactobacillus plantarum directly? The relevance of Lactobacillus plantarum to HSCT is not reflected in the abstract.
-
2、Introduction: Although the authors describe the relationship between the gut microbiome and HSCT, the connection between Lactobacillus plantarum and the gut microbiome is not clarified. The sequence "MDROs – Gut Microbiome – Probiotics – Lactobacillus plantarum" is not well explained, and the overall narrative lacks logic.
-
3、In the results graph of "relative quantification", the issue with the graph bars, which exceed the chart's boundaries, remains unaddressed. The authors have not directly answered the posed questions. Please provide a comprehensive results graph.
-
Author Response
The authors have made many satisfactory amendments to the original study, greatly enhancing its readability. However, there are still significant flaws:
- 1、Abstract: The study background mentions that intestinal colonization by MDROs is a risk factor for early bloodstream infections in HSCT, yet the research aim is "to evaluate the impact of Lactobacillus plantarum on the gut microbiome patterns of HSCT patients colonized with MDROs." Why study Lactobacillus plantarum instead of directly investigating MDROs? Why bypass the gut flora to mention Lactobacillus plantarum directly? The relevance of Lactobacillus plantarum to HSCT is not reflected in the abstract.
- We replaced the background by the following phrase, we could not add more information because of the limit of words.
- Background: Lactiplantibacillus plantarum can inhibit the growth of multidrug-resistant organisms(MDROs) and modulate the gut microbiome, however data in HSCT is scarce.
- 2、Introduction: Although the authors describe the relationship between the gut microbiome and HSCT, the connection between Lactobacillus plantarum and the gut microbiome is not clarified. The sequence "MDROs – Gut Microbiome – Probiotics – Lactobacillus plantarum" is not well explained, and the overall narrative lacks logic.
We added this paragraph that described the in vitro action of L. plantarum against gram-positive and gram-negative bacteria.
Studies in vitro have shown that L. plantarum can inhibit gram-positive and gram-negative bacteria due to bacteriocins and that acetic acid concentration can inhibit the growth of carbapenem-resistant K. pneumonia [15-17]. Lactobacillus plantarum can impair K. pneumoniae preformed biofilm and L. plantarum derived extracellular vesicle modulating host responses to vancomycin-resistant Enterococcus faecium (VRE) [18]
We added the following references:
Kim SW , Kang SI, Shin DH, Oh SY, Lee CW, Yang Y, Son YK, Yang HS, Lee BH, An HJ, Jeong IS, Bang WY . Potential of Cell-Free Supernatant from Lactobacillus plantarum NIBR97, Including Novel Bacteriocins, as a Natural Alternative to Chemical Disinfectants. Pharmaceuticals (Basel). . 2020 Sep 23;13(10):266.
Yan R, Lu Y, Wu X, Yu P, Lan P, Wu X, Jiang Y, Li Q, Pi X, Liu W, Zhou J, Yu Y. Anticolonization of Carbapenem- Resistant Klebsiella pneumoniae by Lactobacillus plantarum LP1812 Through Accumulated Acetic Acid in Mice Intestinal. Front Cell Infect Microbiol. 2021 Dec 15;11:804253
Lagrafeuille R, Miquel S , Balestrino D, Vareille-Delarbre M, Chain F, Langella P, Forestier C. Opposing effect of Lactobacillus on in vitro Klebsiella pneumoniae in biofilm and in an in vivo intestinal colonisation model. 1 Benef Microbes. . 2018 Jan 29;9(1):87-100.
Li M, Lee K, Hsu M, Nau G, Mylonakis E, Ramratnam B Lactobacillus-derived extracellular vesicles enhance host immune responses against vancomycin-resistant enterococci. .BMC Microbiol. 2017 Mar 14;17(1):66. doi: 10.1186/s12866-017-0977-7
- 3、In the results graph of "relative quantification", the issue with the graph bars, which exceed the chart's boundaries, remains unaddressed. The authors have not directly answered the posed questions. Please provide a comprehensive results graph.
We fixed the graph bars as recommend and we hope now they are clear.

Reviewer 3 Report
Comments and Suggestions for Authors
The authors improved the manuscript in accordance with the reviewer's suggestions.
Author Response
We added this paragraph that described the in vitro action of L. plantarum against gram-positive and gram-negative bacteria.
Studies in vitro have shown that L. plantarum can inhibit gram-positive and gram-negative bacteria due to bacteriocins and that acetic acid concentration can inhibit the growth of carbapenem-resistant K. pneumonia [15-17]. Lactobacillus plantarum can impair K. pneumoniae preformed biofilm and L. plantarum derived extracellular vesicle modulating host responses to vancomycin-resistant Enterococcus faecium (VRE) [18]
We added the following references:
Kim SW , Kang SI, Shin DH, Oh SY, Lee CW, Yang Y, Son YK, Yang HS, Lee BH, An HJ, Jeong IS, Bang WY . Potential of Cell-Free Supernatant from Lactobacillus plantarum NIBR97, Including Novel Bacteriocins, as a Natural Alternative to Chemical Disinfectants. Pharmaceuticals (Basel). . 2020 Sep 23;13(10):266.
Yan R, Lu Y, Wu X, Yu P, Lan P, Wu X, Jiang Y, Li Q, Pi X, Liu W, Zhou J, Yu Y. Anticolonization of Carbapenem- Resistant Klebsiella pneumoniae by Lactobacillus plantarum LP1812 Through Accumulated Acetic Acid in Mice Intestinal. Front Cell Infect Microbiol. 2021 Dec 15;11:804253
Lagrafeuille R, Miquel S , Balestrino D, Vareille-Delarbre M, Chain F, Langella P, Forestier C. Opposing effect of Lactobacillus on in vitro Klebsiella pneumoniae in biofilm and in an in vivo intestinal colonisation model. 1 Benef Microbes. . 2018 Jan 29;9(1):87-100.
Li M, Lee K, Hsu M, Nau G, Mylonakis E, Ramratnam B Lactobacillus-derived extracellular vesicles enhance host immune responses against vancomycin-resistant enterococci. .BMC Microbiol. 2017 Mar 14;17(1):66. doi: 10.1186/s12866-017-0977-7